# ZeroDiff: Solidified Visual-Semantic Correlation in Zero-Shot Learning

**Zihan Ye**[1,2,6]**, Shreyank N Gowda**[3]**, Shiming Chen**[4]**, Xiaowei Huang**[2]**,**
**Haotian Xu**[1,2]**, Fahad Shahbaz Khan**[4,5]**, Yaochu Jin**[6]**, Kaizhu Huang**[7]**, Xiaobo Jin**[1†]

[1]Xi'an Jiaotong-Liverpool University, [2]University of Liverpool, [3]University of Nottingham,
[4]Mohamed bin Zayed University of Artificial Intelligence, [5]Linköping University,
[6]Westlake University, [7]Duke Kunshan University
`zihhye@outlook.com`

## Abstract

Zero-shot Learning (ZSL) aims to enable classifiers to identify unseen classes. This is typically achieved by generating visual features for unseen classes based on learned visual-semantic correlations from seen classes. However, most current generative approaches heavily rely on having a sufficient number of samples from seen classes. Our study reveals that a scarcity of seen class samples results in a marked decrease in performance across many generative ZSL techniques. We argue, quantify, and empirically demonstrate that this decline is largely attributable to spurious visual-semantic correlations. To address this issue, we introduce ZeroDiff, an innovative generative framework for ZSL that incorporates diffusion mechanisms and contrastive representations to enhance visual-semantic correlations. ZeroDiff comprises three key components: (1) Diffusion augmentation, which naturally transforms limited data into an expanded set of noised data to mitigate generative model overfitting; (2) Supervised-contrastive (SC)-based representations that dynamically characterize each limited sample to support visual feature generation; and (3) Multiple feature discriminators employing a Wasserstein-distance-based mutual learning approach, evaluating generated features from various perspectives, including pre-defined semantics, SC-based representations, and the diffusion process. Extensive experiments on three popular ZSL benchmarks demonstrate that ZeroDiff not only achieves significant improvements over existing ZSL methods but also maintains robust performance even with scarce training data. Our codes are available at https://github.com/FouriYe/ZeroDiff_ICLR25.

## 1 Introduction

Machine learning models have achieved remarkable success in data-intensive applications, largely due to the availability of abundant labeled samples. However, collecting extensive labeled datasets is often time-consuming and expensive, making it unrealistic to assume access to substantial volumes of labeled data. To improve data efficiency for new classes, zero-shot learning (ZSL) (Xian et al., 2018a) offers a promising solution by transferring knowledge from seen classes to unseen classes through the use of pre-defined class semantic knowledge, such as attributes (Chao et al., 2016) and text-based representations (Zhu et al., 2018; Chen & Yeh, 2021). Generative ZSL methods synthesize features for unseen classes by establishing visual-semantic correlations from seen classes, leading to excellent performance. These methods typically leverage some form of adaptations of generative adversarial networks (GANs) (Han et al., 2021; Gowda, 2023; Hou et al., 2024; Gu et al., 2024) to aid the feature generation.

Despite rapid advances in ZSL, it is generally presumed that there is a substantial number of samples available for each seen class. When labeled data for each seen class are limited, it remains uncertain whether ZSL can still perform effectively. This oversight becomes evident as few ZSL methods consider the scenario of limited samples during training (Verma et al., 2020; Gao et al., 2022;

---

[†]Corresponding author.

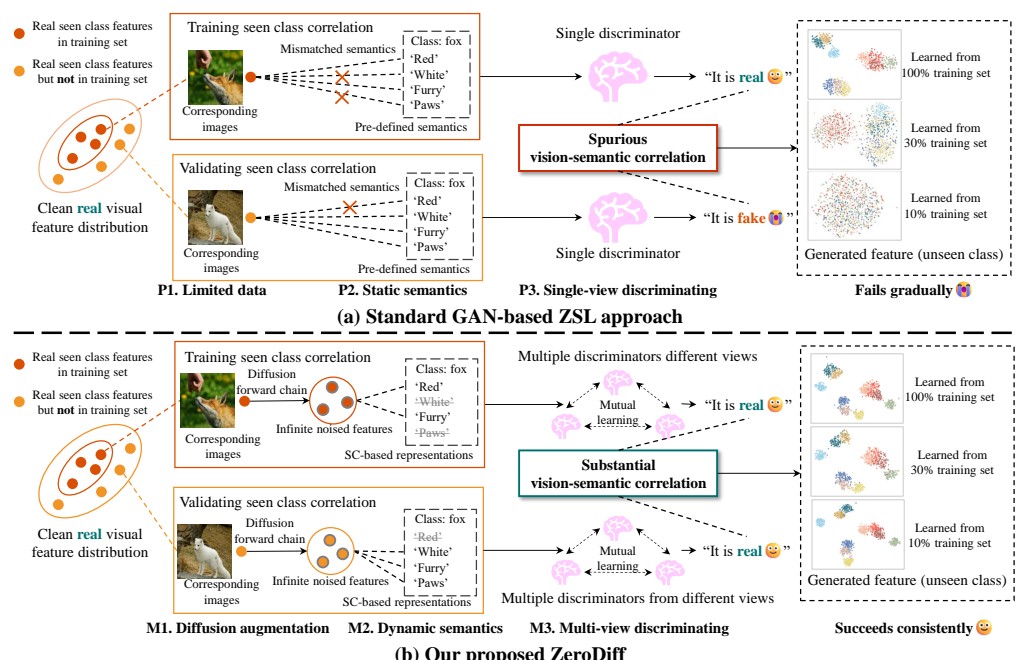

Figure 1: Core idea of our ZeroDiff. (a) Standard GAN-based ZSL approaches suffer from (1) Over-fitting to limited data; (2) Mismatched static pre-defined semantics; (3) Single-view discriminating. Finally, fewer samples in the training set lead to more spurious vision-semantic correlations and feature generation fails gradually. (b) In contrast, the proposed ZeroDiff overcomes these shortcomings using: (1) Diffusion-augmented infinite features; (2) Dynamic SC-based representations; (3) Multi-view discriminating. Finally, ZeroDiff learns substantial vision-semantic correlation and keeps a robust performance with even 10% training set.

Tang et al., 2024). Through empirical investigations, we observe that most existing generative ZSL methods suffer from performance degradation and collapsed generation modes as the number of training samples is gradually reduced. As illustrated in the t-SNE plot in Fig. 1 (a), f-VAEGAN (Xian et al., 2019) fails to synthesize unseen classes effectively when the size of the training set diminishes.

Through further empirical analysis, we find that the degradation might be caused by a **spurious visual-semantic correlation** learned from a limited number of seen samples. As illustrated in Fig. 1 (a), we re-split the real seen class samples into two groups: one for training and the other for validating the correlation. We then feed the visual features of the split samples into the GAN discriminator to obtain critic scores (i.e., the output of the discriminator), which indicate whether the discriminator perceives the input features as real or fake with respect to the class semantics. Next, we calculate the difference in critic scores as a measure to determine whether the learned visual-semantic correlation is spurious or substantial. Our observations reveal that as the number of training samples decreases, the critic score difference becomes increasingly larger, suggesting that the generative models perceive the validating seen class samples as progressively more 'fake.' This phenomenon indicates that a limited number of training samples amplifies the spurious visual-semantic correlation. Further details can be found in Fig. 2.

To strengthen the vision-semantic correlation under conditions of limited seen class samples, we propose a novel ZSL framework: **ZeroDiff**. As shown in Fig. 1 (b), our approach is motivated by three key aspects: (1) **Diffusion augmentation**: Limited training samples can be easily memorized by models. We incorporate the diffusion mechanism (Song et al., 2020a; Wang et al., 2023a) into our method, which allows a single clean sample to be augmented into an infinite number of noised samples by varying the noise-to-data ratios. (2) **Dynamic semantics**: Predefined semantics are static, meaning that class semantics remain the same across different instances. However, each limited sample may only represent part of the predefined semantics. For example, in the AWA2 dataset, all images of the 'fox' class are labeled with the semantics 'red' and 'white', even though this is inaccurate for white foxes. To address this, we revisit the classical Supervised Contrastive (SC) loss (Khosla et al., 2020) and suggest that SC-based representations can generate instance-level semantics for every sample,

enhancing the generation of visual features. (3) **Multi-view discriminating**: We combine three types of discriminators to assess the authenticity of generated features from different perspectives: predefined semantics, the diffusion process, and SC-based representations. To integrate knowledge from all discriminators, we propose a mutual learning loss based on Wasserstein distance, further reinforcing substantial correlations.

To summarize, our contributions are as follows:

- We reveal and quantify the spurious visual-semantic correlation problem, and empirically demonstrate that the problem would be amplified by fewer training samples.
- We propose a novel generative ZSL framework, ZeroDiff, which advances more efficient ZSL in scenarios with limited seen class samples. It strengthens the visual-semantic correlation through diffusion augmentation, dynamic representation of limited samples, and multi-view discrimination.
- We introduce a new protocol to evaluate generative ZSL methods under varying data conditions. Experimental results show that ZeroDiff outperforms various generative models across different amounts of training samples.

## 2 RELATED WORK

### 2.1 ZERO-SHOT LEARNING

ZSL is a research area focused on class-level generalizability (Xian et al., 2018a). The primary approaches to ZSL can be categorized into embedding and generative methods. Embedding methods (Akata et al., 2015; Yang et al., 2016; Ding et al., 2017; Chen et al., 2022b; Ye et al., 2023; Zhang et al., 2025) learn a direct mapping from visual to semantic spaces or vice versa. For example, TransZero++ (Chen et al., 2022a) presents a cross-modal transformer-based architecture to ZSL. However, when training samples are limited, embedding methods often underperform compared to generative methods on smaller datasets(Chen et al., 2022b; Ye et al., 2023).

In contrast, generative ZSL methods use various generative models (Peng et al., 2025) to synthesize visual features for unseen classes and then train a final classifier for these classes. This paper distinguishes itself from previous works in four key aspects: (1) Recent generative ZSL methods have also identified incorrect visual-semantic correlations (Ye et al., 2021; Chen et al., 2024b), but they lack in-depth quantitative analysis. In contrast, we verify this issue using discriminator scores and demonstrate that the problem is exacerbated by a limited number of training samples. (2) Some ZSL works combine GANs and VAEs to address the well-known mode collapse issue (Luo & Yang, 2024), such as f-VAEGAN (Xian et al., 2019), TF-VAEGAN (Narayan et al., 2020), and Bi-VAEGAN (Wang et al., 2023c). However, we show that VAEGAN-based methods still experience mode collapse when the training set is reduced. As a solution, we propose integrating the diffusion mechanism (Ho et al., 2020; Han et al., 2024) to mitigate this problem. (3) To address the limited discriminative power of predefined semantics, recent works propose dynamically updating these semantics, like DSP (Chen et al., 2023) and VADS (Hou et al., 2024). Our approach revisits the classical SC learning (Khosla et al., 2020) and argues that SC-based representations can serve as a new source for instance-level semantics due to their high intra-class variation (Islam et al., 2021). (4) Recent studies (Clark & Jaini, 2024; Li et al., 2023; Rombach et al., 2022) have shown that specific large-scale diffusion models, like Stable Diffusion, also possess zero-shot classification abilities. However, they do not strictly ensure that unseen classes are excluded from training and rely on large-scale models with huge parameters and extensive training sets. In contrast, our interpretation of 'diffusion' stays true to its core principle: a generative paradigm that learns data distributions by denoising noised data. More importantly, our method **does not** violate the ZSL premise (Xian et al., 2018a).

### 2.2 DATA-EFFICIENT GENERATIVE MODELS

Exploring data-efficient generative models is crucial, as the success of generative methods often depends on collecting a vast amount of diverse training samples, which is both costly and challenging (Webster et al., 2019; Saha et al., 2022). In previous works, DiffAugment (Zhao et al., 2020) introduced differentiable augmentation for GANs and successfully trained with only 10% of the data. AdvAug (Chen et al., 2021a) demonstrated that a specific GAN architecture could reduce the amount

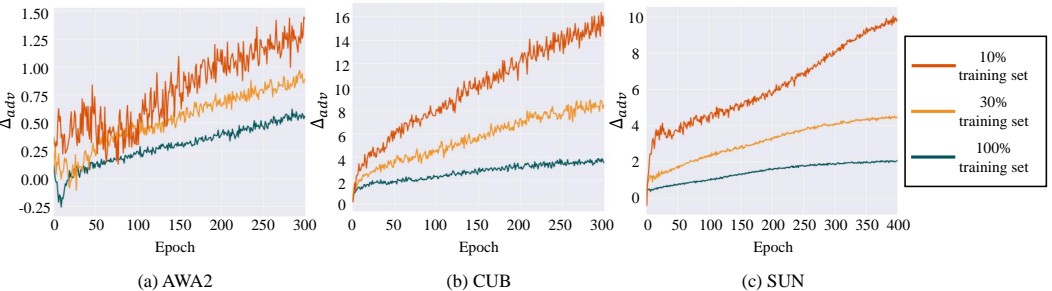

(a) AWA2         (b) CUB         (c) SUN

Figure 2: The $\Delta_{adv}$-epoch curve for the classical f-VAEGAN (Xian et al., 2018b). Larger $\Delta_{adv}$ indicates $D_{adv}$ thinks real testing seen examples are more fake, i.e., learns more spurious visual-semantic correlation.

of required training data using the lottery ticket hypothesis. PatchDiff (Wang et al., 2023a) developed a new conditional score function at the patch level. Denoising Diffusion GAN (DDGAN) (Xiao et al., 2022) suggested that combining diffusion models with GANs could reduce overfitting, though without sufficient empirical validation. Generative ZSL methods can also be seen as a form of data-efficient generative models, as they eliminate the need to collect data or train models for unseen classes. This paper contributes in three key ways: (1) Unlike existing data-efficient generative models, we take it a step further by reducing the need for unseen class data collection and requiring fewer examples from seen classes. (2) We propose quantitative metrics to assess overfitting in visual-semantic correlation. (3) We further integrate GANs and diffusion models using a Wasserstein-distance-based mutual learning approach to distill knowledge across multiple discriminators.

## 3 METHODOLOGY

### 3.1 NOTATIONS

In the ZSL setting, there are two disjoint label sets: a seen set $\mathcal{Y}^s$ used for training and an unseen set $\mathcal{Y}^u$ used for testing, where $\mathcal{Y}^s \cap \mathcal{Y}^u = \emptyset$. The training dataset is denoted as $\mathcal{D}^{tr} = \{(\mathbf{x}^s, y^s, \mathbf{a}^s) \mid \mathbf{x}^s \in \mathcal{X}^s, y^s \in \mathcal{Y}^s, \mathbf{a}^s \in \mathcal{A}^s\}$, where $\mathcal{X}^s$, $\mathcal{A}^s$, and $\mathcal{Y}^s$ represent the image, semantic, and label spaces for the seen classes. The goal of ZSL is to use the training dataset $\mathcal{D}^{tr}$ to create a classifier that can classify unseen images in the testing dataset $\mathcal{D}^{te} = \mathcal{D}^u = \{(\mathbf{x}^u, y^u, \mathbf{a}^u) \mid x^u \in \mathcal{X}^u, y^u \in \mathcal{Y}^u, \mathbf{a}^u \in \mathcal{A}^u\}$, i.e., $f_{zsl} : \mathcal{X}^u \rightarrow \mathcal{Y}^u$. In the Generalized ZSL (GZSL) task, images from seen classes must also be classified during testing. Therefore, a portion of the seen class samples is reserved for testing, denoted as $\mathcal{D}^{te,s}$. In other words, the testing dataset becomes $\mathcal{D}^{te} = \mathcal{D}^{te,s} \cup \mathcal{D}^u$, and the goal becomes $f_{gzsl} : \mathcal{X}^s \cup \mathcal{X}^u \rightarrow \mathcal{Y}^u \cup \mathcal{Y}^s$.

### 3.2 SPURIOUS VISION-SEMANTIC CORRELATION

Given an image $\mathbf{x}^s$ from the training set $\mathcal{D}^{tr}$, existing GAN-based ZSL works use a feature extractor $F_{ce}^*$ pre-trained by Cross-Entropy (CE) loss (Eq. 17) to extract its visual features $\mathbf{v}_0^s = F_{ce}^*(\mathbf{x}^s)$. Then, a feature generator $G$ takes class semantics $\mathbf{a}^s$ and latent variables $\mathbf{z}$ as inputs to synthesize class-specific sample features $\tilde{\mathbf{v}}_0^s = G_{adv}(\mathbf{a}^s, \mathbf{z})$; and a discriminator $D_{adv}$ is used to distinguish real features $\mathbf{v}_0^s$ from fake features $\tilde{\mathbf{v}}_0^s$ by predicting the Wasserstein distance $W_{adv}$ between them according to predefined semantics $\mathbf{a}^s$. Specifically, take the baseline f-VAEGAN as an example, its $D_{adv}$ maximizes the following loss $\mathcal{L}_{adv}$:

$$\mathcal{L}_{adv} = W_{adv} - \lambda_{gpadv}\mathcal{L}_{gpadv}, \tag{1}$$

$$W_{adv} = \mathbb{E}[D_{adv}(\mathbf{v}_0^s, \mathbf{a}^s)] - \mathbb{E}[D_{adv}(\tilde{\mathbf{v}}_0^s, \mathbf{a}^s)], \tag{2}$$

$$\mathcal{L}_{gpadv} = \mathbb{E}[(\|\nabla_{\hat{\mathbf{v}}_0^s} D_{adv}(\hat{\mathbf{v}}_0^s, \mathbf{a}^s)\|_2 - 1)^2], \tag{3}$$

where $\hat{\mathbf{v}}_0^s = \alpha\mathbf{v}_0^s + (1-\alpha)\tilde{\mathbf{v}}_0^s$ with $\alpha \sim U(0,1)$, and $\lambda_{gpadv}$ is a coefficient of the gradient penalty term $\mathcal{L}_{gpadv}$ (Gulrajani et al., 2017) that aims to stabilize the training of GANs.

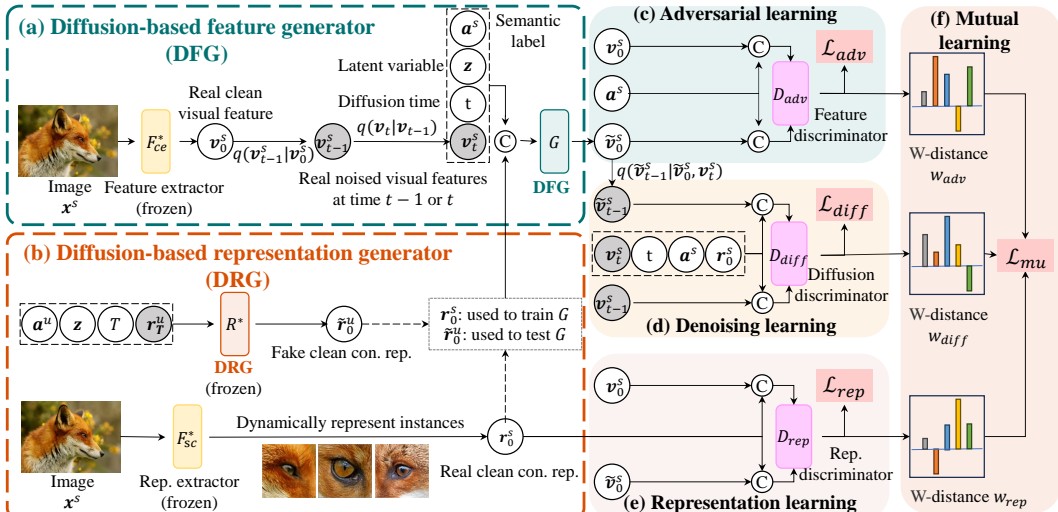

Figure 3: Training pipeline of our DFG. ⓒ represents the concatenation operation. Given frozen extractors $F_{ce}^*$ and $F_{sc}^*$, we take them to extract clean visual features $\mathbf{v}_0$ and contrastive representations $\mathbf{r}_0$. Then, we use the diffusion forward chain (Eq. 5) to obtain real noised visual features $\mathbf{v}_{t-1}$ and $\mathbf{v}_t$. Next, $G$ in DFG denoises/generates a fake clean feature $\tilde{\mathbf{v}}_0$, conditioned by the concatenation of the semantic label $\mathbf{a}$, latent variable $\mathbf{z}$, diffusion time $t$, noised feature $\mathbf{v}_t$, and SC-based representation $\mathbf{r}_0$. The fake clean feature is evaluated from three different learning perspectives: adversarial learning (*Does it match predefined semantics?*), denoising learning ( *Does it match diffusion processes?*), and representation learning (*Does it match contrastive representations?*). Finally, we present the mutual learning loss $\mathcal{L}_{mu}$ to integrate knowledge of all discriminators.

Since a higher critic score means that discriminators consider the input data more 'real', we take the critic score difference between real training features $\mathbf{v}_0^s$ and testing seen class features $\mathbf{v}_0^{te,s}$ to indicate whether the learned vision-semantic correlation is spurious or substantial, i.e.,

$$\Delta_{adv}(\mathbf{v}_0^s, \mathbf{v}_0^{te,s}) = D_{adv}(\mathbf{v}_0^s, \mathbf{a}^s) - D_{adv}(\mathbf{v}_0^{te,s}, \mathbf{a}^{te,s}). \tag{4}$$

Taking this metric, we display the results of f-VAEGAN in Fig. 2. We can find $\Delta_{adv}$ continuously increases and the difference in critic score becomes more significant on smaller training sets, showing that the learned vision-semantic correlation is spurious; this explains why the performances of the generative ZSL methods could drop when the training set shrinks.

## 3.3 ZERODIFF

To solidify the visual-semantic correlation, we propose our ZeroDiff based on our three key insights. The entire training and testing algorithms can be found in Appendix A.1. We now gradually introduce our proposed components that are motivated by our three insights. These components are integrated into the baseline f-VAEGAN resulting in the proposed ZeroDiff.

### 3.3.1 DIFFUSION AUGMENTATION

**Discriminator Overfitting** Following traditional GAN-based methods, we also adapt $D_{adv}$ and the traditional adversarial loss (Eq. 1) to determine whether the generated features align with the predefined semantics, as shown in Fig. 3(c). However, one of the causes of spurious visual-semantic correlation is that limited training sets are memorized by $D_{adv}$. To this end, the first key insight is to leverage the diffusion mechanism to augment the limited training set into infinite noised data to mitigate the overfitting. Specifically, we propose our Diffusion-based Feature Generator (DFG) $G$, as shown in Fig. 3 (a).

**Diffusion-based Generating** Given an image $\mathbf{x}^s$ from the training set $\mathcal{D}^{tr}$, we use $F_{ce}^*$ and $F_{sc}^*$ to extract its clean visual features $\mathbf{v}_0^s = F_{ce}^*(\mathbf{x}^s)$ and clean contrastive representation $\mathbf{r}_0^s = F_{sc}^*(\mathbf{x}^s)$

(The $F_{sc}^*$ and $\mathbf{r}_0^s$ are introduced in the next section). Following previous diffusion models (Ho et al., 2020; Xiao et al., 2022), we apply the diffusion noising process to generate infinite data at various noise levels, from weak to strong, according to the diffusion forward chain:

$$q(\mathbf{v}_{1:T}^s|\mathbf{v}_0^s) = \prod_{t \geq 1} q(\mathbf{v}_t^s|\mathbf{v}_{t-1}^s), \tag{5}$$

$$q(\mathbf{v}_t^s|\mathbf{v}_{t-1}^s) = \mathcal{N}(\mathbf{v}_t^s; \sqrt{1-\beta_t}\mathbf{v}_{t-1}^s, \beta_t\boldsymbol{I}), \tag{6}$$

where $\beta_t$ is a pre-defined variance schedule. When $t = T$, the noised feature $\mathbf{v}_t^s$ become fully Gaussian noise. We set the maximum diffusion time as $T$ and randomly sample a diffusion time $t \sim U(1, \cdots, T)$. We denote the noised visual features as $\mathbf{v}_{t-1}^s$ and $\mathbf{v}_t^s$ at the diffusion times $t-1$ and $t$, respectively. Next, to model the denoising process, we concatenate the noised feature $\mathbf{v}_t^s$, diffusion time $t$, predefined class semantics $\mathbf{a}^s$, latent variable $\mathbf{z}$, and contrastive representation $\mathbf{r}_0$ as the input of $G$, i.e. $\tilde{\mathbf{v}}_0^s = G(\mathbf{a}^s, \mathbf{r}_0^s, t, \mathbf{v}_t^s, \mathbf{z})$. After denoising/generating a clean feature $\tilde{\mathbf{v}}_0^s$, it is evaluated from three aspects: adversarial learning, denoising learning, and representation learning.

**Diffusion-based Discriminating** To evaluate whether the denoised/generated clean features align with the diffusion processes, we design the diffusion discriminator $D_{diff}$, as illustrated in Fig. 3(d). $D_{diff}$ needs to approximate the true denoising distribution $q(\mathbf{v}_{t-1}^s|\mathbf{v}_t^s)$. To this end, we posterior-sample the real noised visual feature $\mathbf{v}_{t-1}^s$ as well as the fake noised visual feature $\tilde{\mathbf{v}}_{t-1}^s$:

$$\tilde{\mathbf{v}}_{t-1}^s \sim q(\tilde{\mathbf{v}}^s{}_{t-1}|\tilde{\mathbf{v}}_0^s, \mathbf{v}_t^s) = \mathcal{N}(\tilde{\mathbf{v}}_{t-1}^s; \tilde{\mu}_t(\mathbf{v}_t^s, \tilde{\mathbf{v}}_0^s), \tilde{\beta}_t\boldsymbol{I}), \tag{7}$$

$$\tilde{\mu}_t(\mathbf{v}_t^s, \mathbf{v}_0^s) = \frac{\sqrt{\bar{\alpha}_{t-1}}\beta_t}{1-\bar{\alpha}_t}\mathbf{v}_0^s + \frac{\sqrt{\alpha_t}(1-\bar{\alpha}_{t-1})}{1-\bar{\alpha}_t}\mathbf{v}_t^s, \tag{8}$$

where $\tilde{\beta}_t = \frac{1-\bar{\alpha}_{t-1}}{1-\bar{\alpha}_t}\beta_t$ and $\bar{\alpha}_t = \prod_{j=1}^t(1-\beta_j)$. Then, we train $D_{diff}$ to learn the Wasserstein distance between them:

$$\mathcal{L}_{diff} = W_{diff} - \lambda_{gpdiff}\mathcal{L}_{gpdiff}, \tag{9}$$

$$W_{diff} = \mathbb{E}[D_{diff}(\mathbf{v}_{t-1}, \mathbf{v}_t, \mathbf{r}_0, \mathbf{a}, t)] - \mathbb{E}[D_{diff}(\tilde{\mathbf{v}}_{t-1}, \mathbf{v}_t, \mathbf{r}_0, \mathbf{a}, t)], \tag{10}$$

$$\mathcal{L}_{gpdiff} = \mathbb{E}[(\|\nabla_{\hat{\mathbf{v}}_{t-1}^s}D_{diff}(\hat{\mathbf{v}}_{t-1}^s, \mathbf{v}_t^s, \mathbf{r}_0^s, \mathbf{a}^s, t)\|_2 - 1)^2], \tag{11}$$

where $\hat{\mathbf{v}}_{t-1}^s = \alpha\mathbf{v}_{t-1}^s + (1-\alpha)\tilde{\mathbf{v}}_{t-1}^s$ with $\alpha \sim U(0, 1)$. Our $G$ minimizes $\mathcal{L}_{gpdiff}$, which equates to minimizing the learned divergence per denoising step:

$$\sum_{t \geq 1} \mathbb{E}[D_{diff}(q(\mathbf{v}_{t-1}^s|\mathbf{v}_t^s)) \| p_G(q(\tilde{\mathbf{v}}_{t-1}^s|\mathbf{v}_t^s))]. \tag{12}$$

### 3.3.2 SC-BASED REPRESENTATIONS

**Static Class-level Semantics** Another cause of spurious visual-semantic correlation is the use of class-level semantics. In other words, existing semantic labels $\mathbf{a}$ are class-level, meaning all instances within a class share the same semantic label. This can result in mismatched correlations between instances and their semantics, as each limited sample may only represent a subset of the predefined semantics. For example, as shown in Fig. 1, all images in the "fox" class are labeled with the semantics "red" and "white," even though white foxes are not "red." Such mismatches further amplify spurious visual-semantic correlations.

**SC-based Representations** To address the static class semantic problem, we revisit the SC loss (Khosla et al., 2020) and point out that SC-based representations could be used to represent instance-level semantics. Previous work (Islam et al., 2021) showed that SC-based representations have larger inter-class variation than those of CE-based features. This indicates that SC-based representations mirror the characteristics for every instance within classes. Our empirical study in Appendix A.4 also verifies this point (Fig. 6 and Fig. 7). For example, the different sub-classes of fox, e.g., white fox, red fox and grey fox, are clustered in the contrastive space as shown in Fig. 7. Thus, except fine-tuning the CE-based extractor $F_{ce}$, we also fine-tune $F_{sc}$ with the SC loss (Eq. 18), and fix it as $F_{sc}^*$ to extract contrastive representations $\mathbf{r}_0^s = F_{sc}^*(\mathbf{x}^s)$. The extracted $\mathbf{r}_0^s$ is taken as the input to $G$ for instance-level semantics, as shown in Fig. 3(b).

**Representation Discriminating** As shown in Fig. 3(e), the representation discriminator $D_{rep}$ is responsible for distinguishing features via the contrastive representation view. It operates in a similar manner to $D_{adv}$, but with the pre-defined semantics $\mathbf{a}$ replaced by the contrastive representation:

$$\mathcal{L}_{rep} = W_{rep} - \lambda_{gprep}\mathcal{L}_{gprep}, \tag{13}$$

where $W_{rep}$ and $\mathcal{L}_{gprep}$ is in Eq. 19 and Eq. 20.

**Unseen Representation Generating** At the testing stage, we cannot get real unseen class representations $r_0^u$ and feed to $G$. Thus, we train another representation generator DRG $R$ to learning the mapping between instance-level SC representations $r_0$ from class-level semantic labels $a$. The DRG training algorithm is similar to DFG, provided in Alg. 2.

### 3.3.3 MUTUAL-LEARNED DISCRIMINATORS

As shown in Fig. 3(f), since our three discriminators evaluate features in different ways, they have different criteria for judging them. If we can enable them to learn mutually, we can obtain stronger discriminators, resulting in better guidance for the generator. For example, the objectives of $D_{adv}$ and $D_{diff}$ are two very similar but distinct tasks: one separates clean features and the other separates noised features. Clearly, separating noised features is a harder task because with more diffusion steps, less information remains. Thus, if we can distill the knowledge from $D_{adv}$ to $D_{diff}$, we can enhance the separation ability on noised features and use the stronger $D_{diff}$ to improve denoising. In contrast, distilling the knowledge from $D_{diff}$ to $D_{adv}$ could prevent $D_{adv}$ from memorizing training samples. To this end, we propose the Wasserstein-distance-based distillation loss:

$$\mathcal{L}_{mu} = \kappa_t^\gamma * (\|W_{diff} - W_{adv}\|_1 + \|W_{diff} - W_{rep}\|_1) + \|W_{adv} - W_{rep}\|_1, \tag{14}$$

where $\kappa_t$ is the Noise-to-Data (N2D) ratio, represented as $1 - \sqrt{\prod_{j=1}^t (1 - \beta_j)}$, and $\gamma$ is a smoothing factor. As $t$ increases, $\kappa_t$ also increases, making it harder to distinguish between fake and real noised features, and $W_{diff}$ provides less guidance for $W_{rep}$ and $W_{adv}$. Therefore, we introduce a smoothing factor $\gamma \geq 0$ to control the strength of discriminator alignment.

### 3.4 OVERALL OPTIMIZATION AND ZSL INFERENCE

The ZeroDiff model alternately trains $G$ and three discriminators $D_{adv}$, $D_{diff}$, and $D_{rep}$ to optimize the following objective function:

$$\min_G \max_{D_{adv}, D_{diff}, D_{rep}} (\mathcal{L}_{adv} + \mathcal{L}_{diff} + \mathcal{L}_{rep} - \lambda_{mu}\mathcal{L}_{mu}), \tag{15}$$

where $\lambda_{mu}$ is the hyper-parameter related to the mutual learning. After completing the training, we freeze them which are denoted as $R^*$ and $G^*$.

For the final ZSL inference, the test images $\mathcal{X}^u$ are projected to the visual feature space $\mathcal{V}^u$ by the frozen $F_{ce}^*$. Then, we adopt the frozen $R^*$ to sample contrastive representations and feed them into the frozen $G^*$ to synthesize visual features $\tilde{\mathbf{x}}_0^u$ of unseen classes (generating $N_{syn}$ samples for each unseen class). Next, we train the final classifier $F_{zsl}$, i.e. $\mathcal{X}^u \rightarrow \mathcal{V}^u \rightarrow \mathcal{Y}^u$.

## 4 EXPERIMENTS

### 4.1 DATASET

We conduct experiments on three popular ZSL benchmarks: AWA2 (Xian et al., 2018a), CUB (Welinder et al., 2010) and SUN (Patterson & Hays, 2012). AWA2 consists of 65 animal classes with 85-D attributes and includes 37,322 images. CUB is a bird dataset containing 11,788 images of 200 bird species. SUN has 14,340 images of 645 seen and 72 unseen classes of scenes. We follow the commonly used setting (Xian et al., 2018a) to divide the seen and unseen classes. We report the average per-class Top-1 accuracy of unseen classes for ZSL. For GZSL, we evaluate the Top-1 accuracy on seen classes ($S$) and unseen classes ($U$), and report their harmonic mean $H = \frac{2 \times S \times U}{S+U}$.

### 4.2 IMPLEMENTATION DETAILS

The imlementation deatils are provided in Appendix A.2.

Table 1: Comparisons with the state-of-the-arts. $U$, $S$, and $H$ represent the top-1 accuracy (%) of unseen classes, seen classes, and their harmonic mean, respectively. The best and second-best results are marked in **Red** and **Blue**, respectively. † denoted the results using our fine-tune features, while ‡ using other fine-tune features. The upper group indicates embedding methods and the lower group is for generative methods.

| Method | Venue | Backbone | ZSL | | | GZSL | | | | | | | | |
| | | | AWA2 | CUB | SUN | AWA2 | | | CUB | | | SUN | | |
| | | | T1 | T1 | T1 | U | S | H | U | S | H | U | S | H |
| CLIP* | ICML21 | ViT | - | - | - | - | - | - | 55.2 | 54.8 | 55.0 | - | - | - |
| CoOp* | IJCV22 | ViT | - | - | - | - | - | - | 49.2 | 63.8 | 55.6 | - | - | - |
| ICIS | ICCV23 | Res101 | 64.6 | 60.6 | 51.8 | 35.6 | **93.3** | 51.6 | 45.8 | 73.7 | 56.5 | 45.2 | 25.6 | 32.7 |
| ReZSL | TIP23 | Res101 | 70.9 | 80.9 | 63.0 | 63.8 | 85.6 | 73.1 | 72.8 | 74.8 | 73.8 | 47.4 | 34.8 | 40.1 |
| PSVMA | CVPR23 | ViT | - | - | - | 73.6 | 77.3 | 75.4 | 70.1 | 77.8 | 73.8 | 61.7 | 45.3 | 52.3 |
| ZSLViT | CVPR24 | ViT | 70.7 | 78.9 | 68.3 | 66.1 | 84.6 | 74.2 | 69.4 | 78.2 | 73.6 | 45.9 | 48.4 | 47.3 |
| f-CLSWGAN† | CVPR18 | Res101 | 75.9 | 84.5 | 75.5 | 65.1 | 68.9 | 66.9 | 76.4 | **83.3** | 79.7 | **63.8** | 55.7 | 59.5 |
| f-VAEGAN† | CVPR19 | Res101 | 75.8 | 85.1 | 75.4 | 67.3 | 65.6 | 66.4 | 77.4 | **83.5** | 80.3 | 63.6 | 54.1 | 58.4 |
| TFVAEGAN† | ECCV20 | Res101 | 72.4 | 85.8 | 74.1 | 54.2 | **89.6** | 67.5 | 79.0 | **83.3** | **81.1** | 51.8 | 53.8 | 52.8 |
| SDVAE† | ICCV21 | Res101 | 69.3 | 85.1 | 77.0 | 57.0 | 72.3 | 63.8 | **81.6** | 74.2 | 77.7 | 62.3 | 56.9 | 59.5 |
| CEGAN† | CVPR21 | Res101 | 72.8 | 84.6 | 74.1 | 57.1 | 89.0 | 69.6 | 78.9 | 80.9 | 79.9 | 58.1 | 57.4 | 57.8 |
| DFCAFlow† | TCSVT23 | Res101 | 74.4 | 83.9 | **77.2** | 67.6 | 81.0 | 73.7 | 77.3 | 82.9 | 80.0 | 63.0 | **59.6** | **61.2** |
| CoOp+SHIP*‡ | ICCV23 | ViT | - | - | - | - | - | - | 55.3 | 58.9 | 57.1 | - | - | - |
| DSP‡ | ICML23 | Res101 | - | - | - | 60.0 | 86.0 | 70.7 | 51.4 | 63.8 | 56.9 | 48.3 | 43.0 | 45.5 |
| DML | ACCV24 | Res101 | - | - | - | 62.2 | 82.3 | 70.9 | 57.1 | 81.6 | 67.2 | 39.6 | 52.7 | 45.9 |
| VADS‡ | CVPR24 | ViT | **82.5** | **86.8** | 76.3 | **75.4** | 83.6 | **79.3** | 74.1 | 74.6 | 74.3 | **64.6** | 49.0 | 55.7 |
| **ZeroDiff**† | **Ours** | Res101 | **87.3** | **87.5** | **77.3** | **74.7** | 89.3 | **81.4** | **80.0** | 83.2 | **81.6** | 63.0 | **56.9** | 59.8 |

## 4.3 COMPARISON WITH STATE-OF-THE-ART

We select representative generative ZSL methods across different generative model types. These methods include: 1. **GAN-based methods**: f-CLSWGAN (Xian et al., 2018b) and CEGAN (Han et al., 2021); 2. **VAE-based methods**: SDVAE (Chen et al., 2021b); 3. **VAEGAN-based methods**: f-VAEGAN (Xian et al., 2019), TFVAEGAN (Narayan et al., 2020), DSP (Chen et al., 2023), DML Zhang et al. (2024) and VADS Hou et al. (2024); 4. **Flow-based method**: DFCAFlow (Su et al., 2023). We also provide comparisons with embedding-based SOTAs: ICIS (Christensen et al., 2023), ReZSL (Ye et al., 2023), PSVMA (Liu et al., 2023) and ZSLViT (Chen et al., 2024a). Besides, large-scale vision-language models, i.e. CLIP (Radford et al., 2021), CoOp (Zhou et al., 2022) and SHIP (Wang et al., 2023b), have shown significant potential for dataset-level ZSL ability. But they do not follow the strict class splitting. Nonetheless, we still include them.

The results are presented in Table 1. For ZSL, compared to all embedding and generative methods, our ZeroDiff consistently achieves the best performance of 87.3%, 87.5%, and 77.3% on AWA2, CUB and SUN datasets, respectively. Our ZeroDiff obtains significant performance boost compared to other generative methods, i.e., by 3.9% and 0.7% on AWA2 and CUB, respectively. For GZSL, our ZeroDiff performs even better at AWA2 and CUB. Our $H$ performances are 79.5% and 81.6%, whilst the second best are only 73. 7% and 81. 1%. Notably, our ZeroDiff also significantly outperforms the large-scale vision-language based methods (e.g. CLIP, CoOp and SHIP).

## 4.4 LIMITED TRAINING DATA

To investigate the performance of generative ZSL approaches under limited training data, we randomly keep training samples of each seen class by different ratios. As the remaining ratio decreases, ZSL becomes more challenging since over-fitting to the training samples becomes much easier. For fairness, we decrease the number of synthesized unseen features proportionally to avoid class imbalance in GZSL. The results are reported in Table 2.

The following observations can be made: _First_, GAN-based approaches, f-CLSWGAN and CEGAN, deteriorate severely when only limited training data are available. This indicates that GANs are fragile under limited data conditions. _Second_, f-VAEGAN combines VAE and GAN to alleviate the mode collapse of GAN. However, when the training set is limited, the mode collapse problem still remains,

Table 2: Comparison on limited training data. We evaluate generative methods with 30% and 10% training samples. $T1$ represents the top-1 accuracy (%) of unseen classes in ZSL. In GZSL, $U$, $S$, and $H$ represent the top-1 accuracy (%) of unseen classes, seen classes, and their harmonic mean, respectively. The best and second-best results are marked in **Red** and **Blue**, respectively.

| Method | AWA2 | | | | CUB | | | | SUN | | | |
| | $30\%\mathcal{D}^{tr}$ | | $10\%\mathcal{D}^{tr}$ | | $30\%\mathcal{D}^{tr}$ | | $10\%\mathcal{D}^{tr}$ | | $30\%\mathcal{D}^{tr}$ | | $10\%\mathcal{D}^{tr}$ | |
| | T1 | H | T1 | H | T1 | H | T1 | H | T1 | H | T1 | H |
|---|---|---|---|---|---|---|---|---|---|---|---|---|
| f-CLSWGAN (Xian et al., 2018b) | 68.9 | 57.8 | 54.0 | 35.7 | 82.1 | 75.3 | 75.1 | 66.8 | 73.4 | **51.5** | 66.9 | 29.3 |
| f-VAEGAN (Xian et al., 2019) | **81.2** | 64.9 | 73.1 | 54.4 | 84.5 | **78.3** | **81.5** | **75.0** | 72.0 | 49.0 | 58.4 | 27.8 |
| CEGAN (Han et al., 2021) | 72.2 | 70.4 | 69.0 | 66.3 | 83.6 | 77.6 | 81.3 | 74.8 | 65.9 | 45.6 | - | - |
| DFCAFlow (Su et al., 2023) | 74.5 | **72.6** | **77.9** | **70.7** | 82.7 | 77.2 | 80.3 | 74.1 | **74.2** | 55.8 | **70.2** | **31.3** |
| **ZeroDiff (Ours)** | **84.9** | **80.2** | **83.3** | **77.0** | **85.5** | **78.7** | **82.9** | **76.1** | **75.4** | 51.3 | **68.1** | **33.3** |

which sometimes performs even worse than the methods only using GAN. For example, with 30% $\mathcal{D}^{tr}$ of SUN, f-VAEGAN achieves 72.0% T1 performance but the vanilla f-CLSWGAN achieves 73.4% T1. *Third*, with the introduction of the diffusion mechanism and instance-level representations, our ZeroDiff achieves the best performance in most cases. This demonstrates the effectiveness of our method in improving data efficiency.

We also provide a qualitative comparison between the baseline f-VAEGAN and our ZeroDiff using t-SNE visualization of the real and synthesized sample features in Fig. 11. The visualization shows that as the number of training samples decreases, f-VAEGAN gradually fails to generate unseen classes, while our ZeroDiff maintains a highly robust ability to generate them.

## 4.5 ABLATION STUDY

We conduct the component effectiveness study in Sec. 4.5.1, hyper-parameter sensitivity in Sec. 4.5.2. Besides, we also provide the ablation study about DFG inputs in Appendix. A.5.

Table 3: Ablation study of our ZeroDiff. ✓ and × denote yes and no. ○ denotes using corresponding versions for DRG.

| ID | Component | | | | | | AWA2 | | CUB | | SUN | |
| | $G$ | $R$ | $D_{adv}$ | $D_{diff}$ | $D_{rep}$ | $\mathcal{L}_{mu}$ | T1 | H | T1 | H | T1 | H |
|---|---|---|---|---|---|---|---|---|---|---|---|---|
| a | ✓ | × | ✓ | × | × | × | 79.9 | 67.0 | 83.7 | 78.5 | 74.7 | 57.8 |
| b | ✓ | × | ✓ | ✓ | × | × | 82.3 | 72.9 | 84.1 | 79.5 | 75.1 | 58.7 |
| c | ✓ | × | ✓ | ✓ | × | ✓ | 82.7 | 73.3 | 84.7 | 80.9 | 75.4 | 59.0 |
| d | × | ✓ | ○ | ○ | × | × | 83.8 | 67.1 | 78.3 | 66.8 | 71.6 | 47.6 |
| e | ✓ | ✓ | ✓ | × | × | × | 79.3 | 72.8 | 83.4 | 78.5 | 73.6 | 56.2 |
| f | ✓ | ✓ | × | ✓ | × | × | 80.3 | 69.0 | 85.8 | 80.5 | 73.5 | 56.4 |
| g | ✓ | ✓ | ✓ | ✓ | × | × | 82.9 | 73.4 | 84.8 | 79.1 | 74.4 | 57.0 |
| h | ✓ | ✓ | ✓ | ✓ | × | ✓ | 85.2 | 78.3 | 87.2 | 81.2 | 76.8 | 59.4 |
| i | ✓ | ✓ | ✓ | ✓ | ✓ | × | 84.6 | 77.5 | 84.8 | 79.3 | 76.8 | 59.2 |
| j | ✓ | ✓ | ✓ | ✓ | ✓ | ✓ | 87.3 | 81.4 | 87.5 | 81.6 | 77.3 | 59.8 |

### 4.5.1 COMPONENT EFFECTIVENESS

We examine the effect of the proposed components: $G$, $R$, $D_{adv}$, $D_{diff}$, $D_{rep}$, and $\mathcal{L}_{mu}$. The results are reported in Table 3. We observe that $D_{diff}$ and $\mathcal{L}_{mu}$ consistently improve the performance across the three datasets. Another observation is that, in many cases, $G$ benefits from the inclusion of $R$, although $R$ does not perform better than $G$ alone. It also evidences our motivation that SC-based representations could capture instance-level semantics to support feature generation.

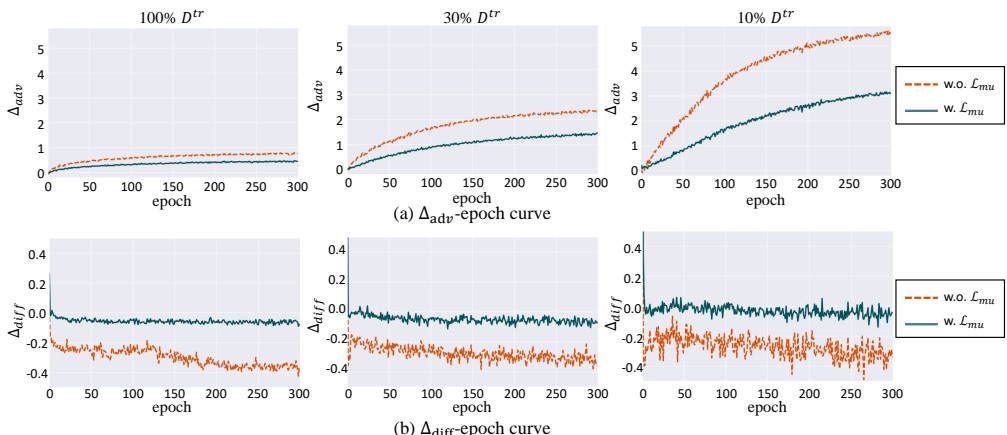

Figure 4: The effect of $\mathcal{L}_{mu}$ to $\Delta_{adv}$ (Eq. 4) and $\Delta_{diff}$ (Eq. 16) on AWA2. (a) indicates that our $\mathcal{L}_{mu}$ mitigates the overfitting of $D_{adv}$ in the training set. (b) shows that the distinguishing ability of $D_{diff}$ is enhanced by our $\mathcal{L}_{mu}$.

### 4.5.2 HYPER-PARAMETER SENSITIVITY

We also provide the hyper-parameter sensitivity analysis for the smoothing factor $\gamma$ (Eq. 14), the loss weight for $\lambda_{mu}$ (Eq. 15) and the number of generated samples per unseen class $N_{syn}$. The results are shown in Fig. 9 and Fig. 10. From Fig. 9 (a), we can find our method performs robust for the change of $\gamma$, and with the N2D reweight, it is consistently better than without N2D reweight. We conclude that we set $\lambda_{mu}$ as 5, and set $\gamma$ as 1.5 on AWA2 to achieve good performance. From Fig. 10, the accuracy for unseen class increases when the number of synthesized samples increases and when $N_{syn}$ is large enough, the performances become stable. This result demonstrates that the features synthesized by our method effectively mitigate the issue of missing data for unseen classes.

### 4.6 MUTUAL LEARNING EFFECTIVENESS

To further verify the effect of our $\mathcal{L}_{mu}$, we designed an additional experiment to show the change in critic score. _First_, as shown in Fig. 4 (a), we display the curve of $D_{adv}$ (Eq. 4). With $\mathcal{L}_{mu}$, $\Delta_{adv}$ decreases significantly compared to the counterpart without $\mathcal{L}_{mu}$ in CUB and SUN, suggesting that knowledge from $D_{diff}$ helps $D_{adv}$ reduces over-fitting in the training set. _Second_, conversely, the knowledge from $D_{adv}$ can also benefit $D_{diff}$. As shown in Fig. 4 (b), we display the critic score difference of $D_{diff}$ between noised real training features $\mathbf{v}_t$ and noised fake training features $\tilde{\mathbf{v}}_t$, i.e.,

$$\Delta_{diff}(\mathbf{v}_t, \tilde{\mathbf{v}}_t) = D_{diff}(\mathbf{v}_t, \mathbf{v}_{t+1}, \mathbf{r}_0, \mathbf{a}, t) - D_{diff}(\tilde{\mathbf{v}}_t, \mathbf{v}_{t+1}, \mathbf{r}_0, \mathbf{a}, t). \tag{16}$$

After adding $\mathcal{L}_{mu}$, $\Delta_{diff}$ increases significantly. This indicates that the distinguishing ability of $D_{diff}$ is enhanced by the knowledge from $D_{adv}$.

## 5 CONCLUSION

In this paper, we investigate the zero-shot learning (ZSL) problem under varying amounts of training data, revealing that the issue of spurious visual-semantic correlation is exacerbated when training samples are scarce. To enhance visual-semantic correlation, we introduce ZeroDiff, which incorporates three key components: (1) a diffusion forward chain to augment the limited training set; (2) SC-based representations to effectively represent each limited sample; and (3) mutually learned discriminators to validate generated features from multiple perspectives, including predefined semantics, contrastive representations, and diffusion processes. Experiments conducted on three popular datasets demonstrate the data efficiency of our method. Our ZeroDiff significantly enhances zero-shot capacity, performing well with both abundant and limited training samples.

ACKNOWLEDGEMENTS

This work is supported by National Natural Science Foundation of China under No. 92370119 and 62376113, Research Development Fund with No. RDF-22-01-02 and No. RDF-TP-0019 and National Natural Science Foundation of China under Grant U1804159.

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

# A  APPENDIX

We present additionally (1) the detailed training and testing algorithm; (2) the detailed implementation details; (3) the detailed comparison of CE-based features and SC-based representations; (4) experimental results of hyper-parameter sensitivity; (5) visualization for generated and real features.

## A.1  FULL ALGORITHM

We include here pseudo-code for training algorithms (Alg.1 and Alg.2) and testing algorithm (Alg. 3). We also provide details of all loss functions here:

**Fine-tuning stage:**

For the feature extractor $F_{ce}$:

$$\mathcal{L}_{CE} = -\sum_{i=1}^{|\mathcal{Y}|} \mathbf{y}_i \log(\widehat{\mathbf{y_i}}). \tag{17}$$

For the representation extractor $F_{sc}$:

$$\mathcal{L}_{SC} = -\log \frac{\exp(\mathbf{h}^\top \mathbf{h}^+/\tau)}{\exp(\mathbf{h}^\top \mathbf{h}^+/\tau) + \sum_K^{k=1} \exp(\mathbf{h}^\top \mathbf{h}_k^-/\tau)}, \tag{18}$$

where $\mathbf{h}^+$, $\mathbf{h}^-$, $K$ and $\tau > 0$ are the positive example (have the same class label with $\mathbf{h}$), negative example (have a different class label to $\mathbf{h}$), the number of $\mathbf{h}^-$ in the batch, a temperature parameter, respectively.

**DFG training:** We supply the $W_{rep}$ and $\mathcal{L}_{gprep}$ here.

$$W_{rep} = \mathbb{E}_{q(\mathbf{v}_0^s)}[D_{rep}(\mathbf{v}_0^s, \mathbf{r}_0^s)] - \mathbb{E}_{p_G(\tilde{\mathbf{v}}_0^s)}[D_{rep}(\tilde{\mathbf{v}}_0^s, \mathbf{r}_0^s)], \tag{19}$$

$$\mathcal{L}_{gprep} = \mathbb{E}_{\substack{q(\mathbf{v}_0^s), \\ p_G(\tilde{\mathbf{v}}_0^s)}}[(\|\nabla_{\hat{\mathbf{v}}_0^s} D_{rep}(\hat{\mathbf{v}}_0^s, \mathbf{r}_0^s)\|_2 - 1)^2]. \tag{20}$$

**DRG training:** Similar to DFG training, our DRG $F$ generates fake clean representation $\tilde{\mathbf{r}}_0^s \leftarrow R(\mathbf{z}, \mathbf{a}^s, \mathbf{r}_{t+1}^s, t)$ by the guidance from an adversarial discriminator $D'_{adv}$ and a diffusion discriminator $D'_{diff}$ to guide SC-based representation generation. The adversarial learning loss of DRG is

$$\mathcal{L}'_{adv} = W'_{adv} - \lambda'_{gpadv}\mathcal{L}'_{gpadv}, \tag{21}$$

$$W'_{adv} = \mathbb{E}[D'_{adv}(\mathbf{r}_0^s, \mathbf{a}^s)] - \mathbb{E}[D'_{adv}(\tilde{\mathbf{r}}_0^s, \mathbf{a}^s)], \tag{22}$$

$$\mathcal{L}'_{gpadv} = \mathbb{E}[(\|\nabla_{\hat{\mathbf{r}}_0^s} D'_{adv}(\hat{\mathbf{r}}_0^s, \mathbf{a}^s)\|_2 - 1)^2]. \tag{23}$$

And the denoising learning loss of DRG is

$$\mathcal{L}'_{diff} = W'_{diff} - \lambda'_{gpdiff}\mathcal{L}'_{gpdiff}, \tag{24}$$

$$W'_{diff} = \mathbb{E}[D'_{diff}(\mathbf{r}_{t-1}^s, \mathbf{r}_t^s, \mathbf{a}^s, t)] - \mathbb{E}[D'_{diff}(\tilde{\mathbf{r}}_{t-1}^s, \mathbf{r}_t^s, \mathbf{a}^s, t)], \tag{25}$$

$$\mathcal{L}'_{gpdiff} = \mathbb{E}[(\|\nabla_{\hat{\mathbf{r}}_{t-1}^s} D_{diff}(\hat{\mathbf{r}}_{t-1}^s, \mathbf{r}_t^s, \mathbf{a}^s, t)\|_2 - 1)^2]. \tag{26}$$

The hyper-parameters $\lambda'_{gpadv}$ and $\lambda'_{gpdiff}$ are set to 10.

---

**Algorithm 1** Fine-Tuning of ZeroDiff

---

**Input**: Training data $\mathcal{D}^{tr} = \{(\mathbf{x}^s, \mathbf{a}^s, y^s) | \mathbf{x}^s \in \mathcal{X}^s, \mathbf{a}^s \in \mathcal{A}^s, y^s \in \mathcal{Y}^s\}$, the iteration number for fine-tuning $N_{ft}$.

**[Fine-tuning with CE and SC]**

Define $F_{ce}$ and $F_{cls}$ as pretrained feature extractor and feature classifier.
Define $F_{sc}$ and $F_{pro}$ as pretrained representation extractor and contrastive projector.
**for** $N_{ft}$ **do**.
    Draw a batch of samples $(\mathbf{x}^s, \mathbf{a}^s, y^s)$ from $\mathcal{D}^{tr}$.
    Extract feature $\mathbf{v}^s \leftarrow F_{ce}(\mathbf{x}^s)$.
    Classify feature $\widehat{y}^s \leftarrow F_{cls}(\mathbf{v}^s)$.
    Update $F_{ce}$ and $F_{cls}$ with $\mathcal{L}_{CE}$ by Eq. 17.
    Extract representation $\mathbf{r}^s \leftarrow F_{sc}(\mathbf{x}^s)$.
    Project representation into contrastive space $\mathbf{h}^s \leftarrow F_{pro}(\mathbf{r}^s)$.
    Update $F_{sc}$ and $H_{sc}$ with $\mathcal{L}_{SC}$ by Eq. 18.
**end for**
Freeze $F_{ce}$ and $F_{sc}$ as $F_{ce}^*$ and $F_{sc}^*$.

**Output**: $F_{ce}^*$ and $F_{sc}^*$.

---

**Algorithm 2** Generator Training of ZeroDiff

---

**Input**: Training data $\mathcal{D}^{tr} = \{(\mathbf{x}^s, \mathbf{a}^s, y^s) | \mathbf{x}^s \in \mathcal{X}^s, \mathbf{a}^s \in \mathcal{A}^s, y^s \in \mathcal{Y}^s\}$, the iteration number for generator training $N_g$, the iteration number for discriminator in a step $N_{dis}$.

**[Training DRG]**

Define $R$ as diffusion-based representation generator.
Define $D'_{adv}$ and $D'_{diff}$ as clean representation discriminator and noised representation discriminator.
**for** $N_g$ **do**
    Draw a batch of samples $(\mathbf{x}^s, \mathbf{a}^s, y^s)$ from $\mathcal{D}^{tr}$.
    **for** $N_{dis}$ **do**
        Extract clean contrastive representation $\mathbf{r}_0^s \leftarrow F_{sc}^*(\mathbf{x}^s)$.
        Sample $t \sim U(0, T-1)$.
        Add noise into clean representation at $t$ and $t+1$ as $\mathbf{r}_t^s$ and $\mathbf{r}_{t+1}^s$ by Eq. 5.
        Generate fake clean representation $\tilde{r}_0^s \leftarrow R(\mathbf{z}, \mathbf{a}^s, \mathbf{r}_{t+1}^s, t), \mathbf{z} \sim \mathcal{N}(0, I)$.
        Renoise fake representation $\tilde{\mathbf{r}}_t^s$ by Eq. 7.
        Update $D'_{adv}$ and $D'_{diff}$ by minimizing Eq. 21 and Eq. 24.
    **end for**
    Update $R$ by maximizing $\mathcal{L}'_{adv}$ and $\mathcal{L}'_{diff}$.
**end for**
Freeze $R$ as $R^*$.

**[Training DFG]**

Define $G$ as diffusion-based feature generator.
Define $D_{adv}$, $D_{diff}$ and $D_{rep}$ as discriminators for clean feature, noised feature and contrastive representation aspects.
**for** $N_g$ **do**
    Draw a batch of samples $(\mathbf{x}^s, \mathbf{a}^s, y^s)$ from $\mathcal{D}^{tr}$.
    **for** $N_{dis}$ **do**
        Extract clean representation $\mathbf{r}_0^s \leftarrow F_{sc}^*(\mathbf{x}^s)$ and feature $\mathbf{v}_0^s \leftarrow F_{ce}^*(\mathbf{x}^s)$.
        Sample $t \sim U(0, \cdots, T-1)$.
        Noising feature at $t$ and $t+1$ as $\mathbf{v}_t^s$ and $\mathbf{v}_{t+1}^s$ by Eq. 5.
        Generate fake clean feature $\tilde{\mathbf{v}}_0^s \leftarrow G(\mathbf{z}, \mathbf{a}^s, \mathbf{r}_0^s, \mathbf{v}_{t+1}^s, t), \mathbf{z} \sim \mathcal{N}(0, I)$.
        Renoise fake feature $\tilde{\mathbf{v}}_t^s$ by Eq. 7.
        Update $D_{adv}$, $D_{diff}$ and $D_{rep}$ by minimizing Eq. 1, Eq. 9, Eq. 13 and Eq. 14.
    **end for**
    Update $G$ by maximizing Eq. 1, Eq. 9 and Eq. 13.
**end for**
Freeze $G$ as $G^*$.

**Output**: $G^*$ and $R^*$.

---

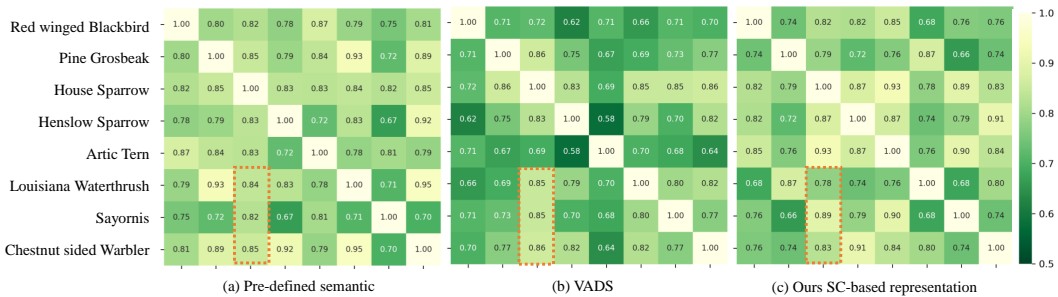

Figure 5: Heatmap comparison of the semantic prototype similarity among (a) pre-defined semantic (Reed et al., 2016a), (b) dynamic VADS (Hou et al., 2024), and (c) our proposed SC-based representation. We randomly select 8 classes on CUB. Our method improves semantic prototypes to distinguish between categories, e.g., the similarities marked by the red dashed line.

---

**Algorithm 3** ZSL Testing of ZeroDiff

**Input**: Testing dataset $\mathcal{D}^{te} = \{(\mathbf{x}^u, \mathbf{a}^u, y^u) | x^u \in \mathcal{X}^u, \mathbf{a}^u \in \mathcal{A}^u, y^u \in \mathcal{Y}^u\}$, trained DFG $F^*$ and DRG $R^*$, the iteration number for pseudo training $N_{te}$, the generation number per class $N_{syn}$.

**[Testing DFG and DRG]**

**for** every unseen class $c^u$ **do**
  Define $\tilde{y}^u$ as copying $c^u$ $N_{syn}$ times and $\tilde{a}^u$ as class semantics.
  Generate fake clean representation $\tilde{\mathbf{r}}_0^s \leftarrow R^*(\mathbf{z}, \mathbf{a}^u, \mathbf{r}_T^u, T-1), \mathbf{z} \sim \mathcal{N}(0, I), \mathbf{r}_T^u \sim \mathcal{N}(0, I)$.
  Generate fake clean feature $\tilde{\mathbf{v}}_0^s \leftarrow G^*(\mathbf{z}, \mathbf{a}^u, \tilde{\mathbf{r}}_0^s, \mathbf{v}_T^u, T-1), \mathbf{z} \sim \mathcal{N}(0, I), \mathbf{v}_T^u \sim \mathcal{N}(0, I)$.
**end for**
Construct a new dataset $\tilde{\mathcal{D}}^{te} = \{(\tilde{v}_0^s, \tilde{r}_0^s, \tilde{a}^u, \tilde{y}^u)\}$.
Define $F_{zsl}$ as the final ZSL classifier.
**for** $N_{te}$ **do**
  Draw a batch of samples $(\tilde{\mathbf{v}}_0^u, \tilde{\mathbf{r}}_0^u, \tilde{\mathbf{a}}^u, \tilde{y}^u)$ from $\tilde{\mathcal{D}}^{te}$.
  Classify pseudo sample $\hat{y}^u \leftarrow F_{zsl}(\tilde{\mathbf{v}}_0^u, \tilde{\mathbf{r}}_0^u)$.
  Update $F_{zsl}$ with $\mathcal{L}_{CE}$ by Eq. 17.
**end for**
Freeze $F_{zsl}$ as $F_{zsl}^*$.
**for** every real $x^u$ **do**
  Extract clean contrastive representation $\mathbf{r}_0^u \leftarrow F_{sc}^*(x^u)$ and clean feature $\mathbf{v}_0^u \leftarrow F_{ce}^*(\mathbf{v}^u)$.
  Classify real sample $\hat{y}^u \leftarrow F_{zsl}^*(\mathbf{v}_0^u, \mathbf{r}_0^u)$.
**end for**

**Output:** ZSL classification results $\{(\hat{y}^u)\}$.

---

## A.2 IMPLEMENTATION DETAILS

For visual features, we extract 2,048-D features for all datasets using ResNet-101 (He et al., 2016) pre-trained with CE on ImageNet-1K (Deng et al., 2009). For contrastive representations, we adopt ResNet-101 pre-trained with PaCo (Cui et al., 2021) on ImageNet-1K. For semantic labels, we use attribute vectors for AWA2 and 1,024-D attributes extracted from textual descriptions (Reed et al., 2016b) for CUB and SUN. We use Adam to optimize all networks with an initial learning rate of 0.0005. For all datasets, $\lambda_{gpadv}$, $\lambda_{gpdiff}$, and $\lambda_{gprep}$ are fixed at 10. Following DDGAN (Xiao et al., 2022), the number of diffusion steps $T$ is set to 4, and we use the discretization of the continuous-time extension, known as the Variance Preserving (VP) SDE (Song et al., 2020b) to compute $\beta_t$ in Eq. 5. All experiments are conducted in Quadro RTX 8000.

## A.3 VADS V.S. SC-BASED REPRESENTATIONS

We find that our SC-based representations exhibit better class discriminability compared to pre-defined semantics and the VADS method, which dynamically updates class-level semantics to instance-level

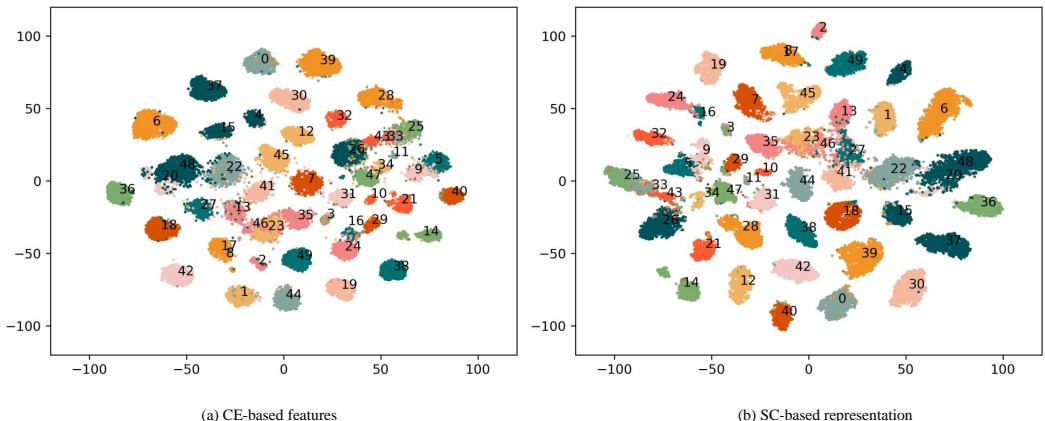

(a) CE-based features      (b) SC-based representation

Figure 6: Comparison CS-based features and SC-based representations with t-SNE visualization for all classes on AWA2. We can find SC-based representations have larger intra-class variation than CE-based features.

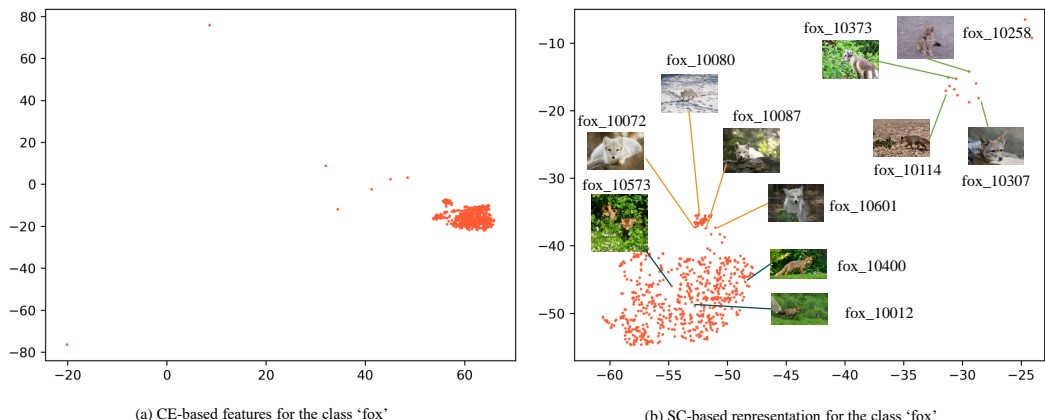

(a) CE-based features for the class 'fox'      (b) SC-based representation for the class 'fox'

Figure 7: Comparison CS-based features and SC-based representations with t-SNE visualization for the class 'fox' on AWA2. We can find all instances of fox are clustered in a group in CE-based space while different sub-classes (i.e. red fox, white fox and grey fox) are gathered in different groups in SC-based space.

semantics. While VADS can also extract instance-level semantics, it occasionally sacrifices class discriminability.

Specifically, as shown in Fig. 5, we randomly select 8 classes from the CUB dataset and visualize the heatmap of class prototype similarities. For the three class similarities marked by the red dash, the similarities from pre-defined class-level semantics are (0.84, 0.82, 0.85), while the similarities produced by VADS remain very similar: (0.85, 0.85, 0.86). In contrast, our SC-based representations make these three hard classes more distinctive, with class similarities of (0.78, 0.89, 0.83).

## A.4   CE-BASED FEATURES V.S. SC-BASED REPRESENTATIONS

We provide more detailed comparison between CE-based features and SC-based representations. First, we use t-SNE to visualize the CE-based features and SC-based representations in Fig. 6. We can find every classes cluster more tightly in CE-based space than in SC-based space. It verifies previous work claims SC-based representations have larger intra-class variation. It also means that SC-based representations contain more intra-class uncertainty. To further verify this point, we visualize the class 'fox' in Fig. 7. We can find that all instances are grouped to a single cluster in CE-based space while different sub-classes of fox (i.e. red fox, white fox and grey fox) could be separately grouped

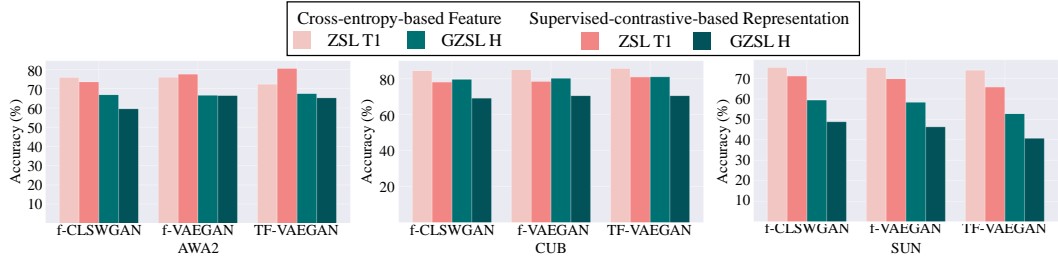

Figure 8: Comparison of performance between CE-based features and SC-based representations. We selected three typical generative ZSL methods to evaluate these two fine-tuning losses. We found that CE is better than SC in most cases, except for the ZSL T1 accuracy in AWA2.

in SC-based space. In other word, SC-based representations could reflect the characteristics of every instances better than CE-based features. Thus, we use SC-based representations as a new source of semantics.

Second, our study also challenge the assumption claimed in previous ZSL that SC-based representations are more discriminative than CE-based features (Han et al., 2021). Recent influential works have shown that SC-based representations are more sensitive to data imbalance (Cui et al., 2021; Jiang et al., 2021; Zhu et al., 2022; Cui et al., 2023; Peng et al., 2024). Fewer training samples lead to poorer SC-trained representations. Thus, we also use classical generative ZSL approaches to evaluate the ZSL performance of SC-based representations and CE-based features. We found that SC performance is generally worse significantly compared to that of traditional CE. The results can be found in Fig. 8.

In short, SC-based representations has better intra-class variations while CE-based features are more discriminative. Thus, directly using SC-based representations to generative ZSL methods is not better than CE-based features, but using SC-based representations as a new semantics is eligible.

## A.5 ABLATION STUDY FOR INPUT OF DFG

Table 4: Ablation study of the input of our DFG. ✓ and × denote yes and no.

| ID | DFG inputs | | | | AWA2 | | CUB | | SUN | |
|----|-----------|------|--------------|-----|------|------|------|------|------|------|
| | $\mathbf{a}$ | $\mathbf{r}_0$ | $\mathbf{v}_t$ | $t$ | T1 | H | T1 | H | T1 | H |
| a | ✓ | × | × | × | 77.0 | 69.8 | 85.5 | 80.0 | 76.8 | 58.3 |
| b | × | ✓ | × | × | 80.0 | 75.7 | 85.1 | 79.9 | 73.1 | 55.9 |
| c | ✓ | ✓ | × | × | 84.3 | 76.0 | 86.9 | 80.6 | 77.0 | 58.5 |
| d | ✓ | ✓ | ✓ | × | 84.2 | 78.1 | 85.5 | 80.3 | 75.0 | 57.6 |
| e | ✓ | ✓ | ✓ | ✓ | 87.3 | 81.4 | 87.5 | 81.6 | 77.3 | 59.8 |

We conduct an ablation study on the inputs of our DFG, as shown in Table 4. The results demonstrate that all inputs are necessary, with the diffusion time $t$ being particularly critical. Specifically: (1) IDs a, b, and c confirm that the class-level semantic label $\mathbf{a}$ and instance-level SC representations $\mathbf{r}_0$ consistently improve performance across the three datasets. (2) IDs c and d show that using only $\mathbf{v}_t$ does not allow the method to fully benefit from reconstructing $\tilde{\mathbf{v}}_0$ from $\mathbf{v}_t$, highlighting the importance of the diffusion time $t$. (3) IDs d and e illustrate that inputting both $\mathbf{v}_t$ and $t$ results in significant improvements, demonstrating the effectiveness of our diffusion augmentation.

## A.6 HYPER-PARAMETER SENSIBILITY

In our ZeroDiff, the main hyper-parameters include the loss weight $\lambda_{mu}$ (Eq. 15), the smoothing factor $\gamma$ that controls the smoothing of the noise-to-data ratio (Eq. 14) and the number of synthesized samples for each unseen class $n_{syn}$. The sensitivity analysis of $\lambda_{mu}$ and $\gamma$ is illustrated in Fig. 9 (a) and (b). The sensitivity varies across different datasets. From Fig. 9 (a), we can find our method

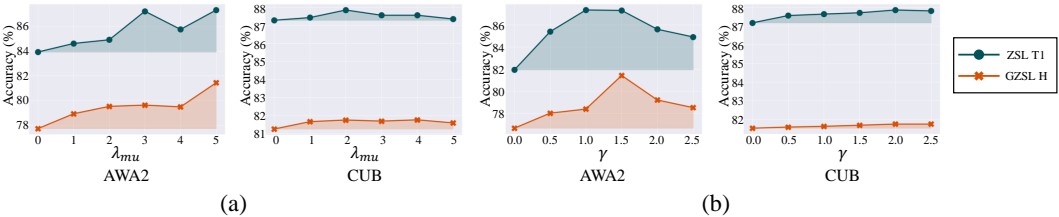

Figure 9: Hyper-parameter Sensitivity. (a) The effect of weight factor $\lambda_{mu}$ (Eq. 15). (b) The effect of smoothing factor $\gamma$ for N2D reweight $\kappa_t^\gamma$ (Eq.14). The shaded area indicates the performance improvement compared to hyper-parameters set as 0.

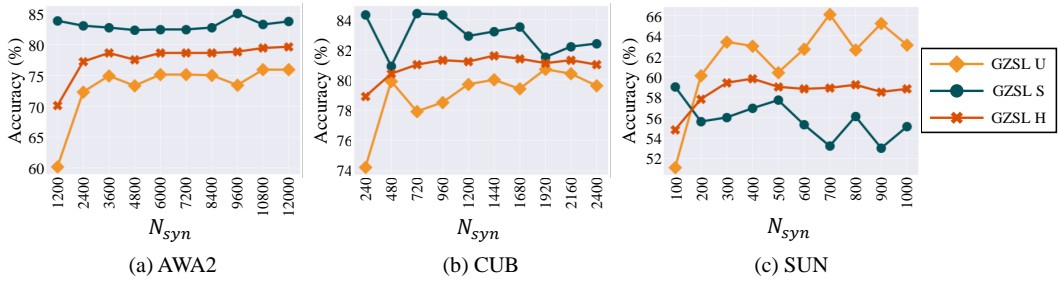

Figure 10: The effect of the number of synthetic samples $N_{syn}$.

perform robust for the change of $\gamma$, and with N2D reweight is consistently better than the counterpart that without N2D reweight. We conclude that we set $\lambda_{mu}$ as 5, and set $\gamma$ as 1.5 on AWA2 to achieve good performance. From Fig. 10, the accuracy for unseen class increases with the number of synthesized samples is up and when $N_{syn}$ is large enough, the performances become stable. This result demonstrates that the features synthesized by our method effectively mitigate the issue of missing data for unseen classes.

## A.7 TSNE VISUALIZATION

We use t-SNE visualization for the real and synthesized sample features to provide a qualitative comparison between the baseline f-VAEGAN and our ZeroDiff in Fig. 11. It is clear that our ZeroDiff maintains a robust generation for unseen classes with scarce seen class samples.

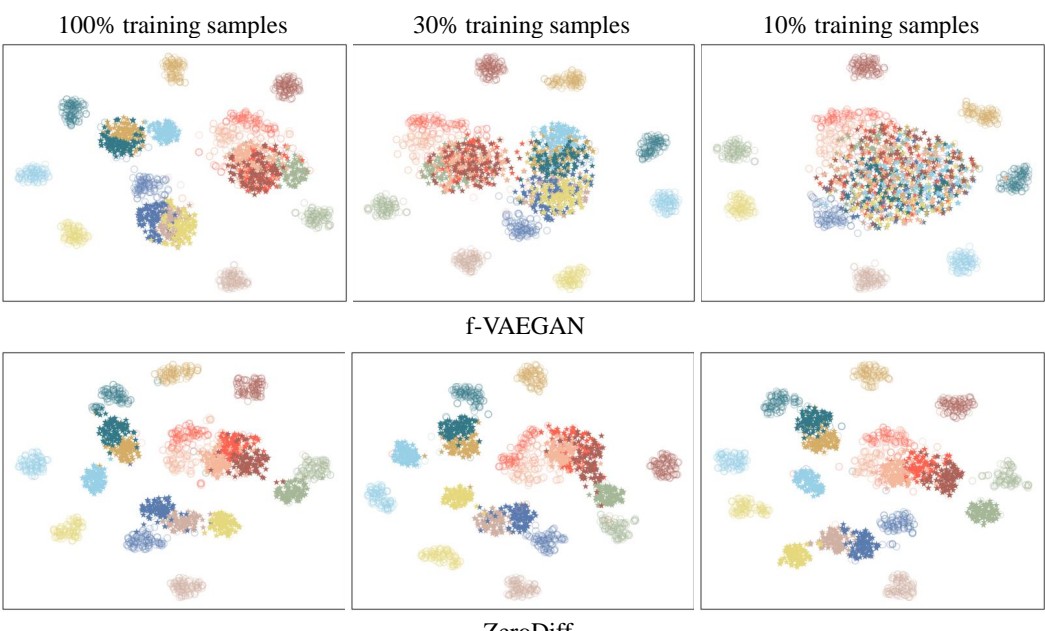

Figure 11: Qualitative evaluation with t-SNE visualization on AWA2. We randomly selected 100 generated samples per class and all real samples for 10 unseen classes. We use different colors to denote classes, and use ∘ and ∗ to denote the real and synthesized sample features, respectively (Best Viewed in Color).

