# OpenReview forum: "ZeroDiff: Solidified Visual-semantic Correlation in Zero-Shot Learning"
_ICLR.cc/2025/Conference — ICLR 2025 Poster_

### Official Review · Reviewer_ZpvL · 2024-10-28

**Soundness:** 3
**Presentation:** 2
**Contribution:** 3
**Rating:** 6
**Confidence:** 5

**Summary:**

The paper proposes a novel diffusion-based generative framework ZeroDiff for ZSL. Starting with a very motivating indicator, the paper claims that current generative methods learn spurious visual-semantic relationships when training data is insufficient, resulting in the failure of generated features. The comparison results well validate the effectiveness of the proposed method.

**Strengths:**

1. The paper proposes a novel diffusion-based generative method for ZSL.
2. The experiments are comprehensive.

**Weaknesses:**

1. The paper claims that generative-based methods learn spurious visual-semantic relationships when training data is insufficient. Is the conclusion applicable to non-generative ZSL methods?
2. As shown in Fig.2, \delta_{adv} increases over the three datasets as the training progresses. However, the paper aims to learn a model with low \delta_{adv} values. Does that mean we are getting a worse model as the training continues? It isn't very clear. The authors need to clarify the relationship between increasing \delta_{adv} values and model performance, and explain how this relates to the goal of learning substantial visual-semantic correlations. It would be helpful to explore what an ideal \delta_{adv} curve should look like.
3. In L262, the authors fine-tune the backbone model by the SC loss. However, the baseline methods do not fine-tune the feature extractor. Therefore, the comparison seems to be unfair. The authors should either apply the same fine-tuning to the baseline methods, or provide results for their method without fine-tuning, to ensure a fair comparison. Additionally, it would be helpful to discuss the impact of this fine-tuning on the overall performance.
4. The whole training process is divided into three stages, which makes it very complicated and hard to read. In addition, the author uses too much information as input for G in L301. Is this reasonable and necessary? The authors need to discuss potential ways to simplify the approach or explain why each component is necessary for the method's performance. It would be helpful to conduct an ablation study that shows the impact of each input on the final performance.

**Questions:**

Please see the weakness above.

---

> ### Author Response · Authors · 2024-11-21
> **Official Comment by Authors for Reviewer ZpvL (Part1)**
>
> Thank you for your valuable feedback! We hope the clarifications below effectively address your concerns.
>
> # W1 (Embedding ZSLs with limited training data):
> Thank you for your suggestion. We use the AttentionNet (Rebalanced Zero-shot Learning, TIP23) to compute the class-averaged Mean-Squared Error (MSE) for semantic prediction.
> The experiments show that semantic errors on test samples from both unseen and seen classes do not increase significantly during training.
> This suggests that spurious visual-semantic correlation might be a unique issue for generative ZSL methods.
>
> At the very least, fewer training samples do not significantly deteriorate semantic prediction in embedding methods.
> However, embedding methods still perform substantially worse, with their performance being up to 16.4\% lower (AWA2 ZSL T1 accuracy) compared to ours.
>
> # W2 (Relationship between $\Delta_{adv}$ and performance):
>
> Figure 2 in the original paper corresponds to the curves from VAEGAN. Our Figure 4 shows the effect of $\mathcal{L}_{mu}$ on the curve.
>
> The $\Delta_{adv}$ is the critic score differences between training real samples and validating real samples.
> If $\Delta_{adv}$ is larger than 0, $D_{adv}$ thinks training real samples are more realistic than validating real samples, vice versa.
> So it can reflect the $D_{adv}$ is over-fitting or under-fitting.
> However, **it is hard to use $\Delta_{adv}$ indicate the ZSL performance.**
> The reasons are mainly from two points:
>
> (a) In generative ZSL methods, the ZSL performance mainly relies on $G$ generation ability.
> If $G$ only can generate poor samples and $D_{adv}$ can easily distinguish fake and real samples, $D_{adv}$ has no need to remember training samples.
> Especially, in the beginning of training (e.g. the first 10 epoches), the ZSL performance is far lower than in the best performance epoch, while the $\Delta_{adv}$ is also low.
>
> (b) In other way, even $\Delta_{adv}$ is very high, $D_{adv}$ still has a basic visual-semantic correlation for training samples, which guarantees a low bound of ZSL performance.
> Thus, when the training goes to late, although $\Delta_{adv}$ is increasing, the performance still does not drop significantly.
>
> The results from our ablation study effectively shows that regularizing  $\Delta_{adv}$ could improve ZSL performance.
> After our regularization term $L_{\text{mu}}$, $\Delta_{adv}$ is more close to the **ideal situation**: should always around 0 during training.
> It indicates that $D_{adv}$ is neither over-fitting nor under-fitting.
>
> # W3 (Experiment Fairness):
>
> We apologize for any confusion and would like to clarify that **our evaluation is fair**. Below, we address this concern from two perspectives:
>
> (1) **Clarification of Results in Table 1 in paper.**
> In Table 1, the results of **all other ZSL generative methods use fine-tuned features**.
> We have used the marker $\dagger$ to denote results obtained using our CE-based features.
> For the remaining methods, DSP and VADS use fine-tuned features from PSVMA, while CoOp+SHIP utilize their own fine-tuned features.
> To avoid similar concerns, we have added the marker $\ddagger$ to denote methods that use other fine-tuned features.
>
> (2) **The Fine-tuning Step as an Indivisible Innovation**.
> Our method distinguishes itself by deeply integrating fine-tuning into the overall approach.
> While previous methods (e.g., f-VAEGAN, TF-VAEGAN) provide results using fine-tuned features, they only replace non-fine-tuned features without exploring how to fine-tune or utilize them effectively in subsequent generation steps.
>
> In contrast, we uncover a key insight: Fine-tuning with SC-loss introduces instance-level semantics.
> This is a critical problem in recent ZSL research (e.g., VADS, DSP), as existing semantic labels are class-level and often misaligned with instances, leading to spurious visual-semantic correlations.
> From domain adaptation studies, it is known that contrastive loss fine-tuned in source domains can capture low- and mid-level visual information (e.g., color).
> Building on this, we hypothesize that it can also capture instance-level semantics.
>
> To verify this, we provide a t-SNE comparison in Fig. 6, showing that SC-based representations have larger intra-class variation than CE-based features.
> This indicates that SC-based representations reflect instance-level variation.
> Further visualization in Fig. 7 reveals that SC-loss can separate the fox class into distinct sub-classes (e.g., red fox, white fox, and grey fox).
> These findings strongly support our hypothesis.
>
> Thus, we use SC-based representations as an additional condition to provide instance-level semantics and mitigate spurious visual-semantic correlations.

---

> > ### Author Response · Authors · 2024-11-21
> > **Official Comment by Authors for Reviewer ZpvL (Part2)**
> >
> > # W4 (Overall Presentation and More Ablation Study):
> >
> >
> > **We have thoroughly refined our manuscript and uploaded the revised PDF, with updates highlighted in red.**
> > Following the suggestions from reviewers FPAJ and KQOZ, we now provide a step-by-step explanation of our three key insights and their connections to the corresponding modules in our methodology.
> > In this version, readers can smoothly transition from the baseline f-VAEGAN to our proposed ZeroDiff.
> > You can review these changes in the revised manuscript PDF and our summarized **Common response 1**.
> > We hope these revisions enhance readability and effectively address your concerns.
> >
> > The ablation study is another important consideration, and we have supplemented additional studies as part of **Common response 2** in two ways:
> >
> > (1) **More Compositions of Modules and Losses**.
> > Following the suggestion from reviewer FPAJ, we added results on additional compositions.
> > In addition to $G+R+D_{\text{adv}}$ and $G+D_{\text{adv}}+D_{\text{diff}}+L_{\mu}$, we included $G+R+D_{\text{diff}}$, $G+R+D_{\text{adv}}+D_{\text{diff}}$, and $G+R+D_{\text{adv}}+D_{\text{diff}}+L_{\mu}$.
> > A notable observation is that adding $R$ to support $G$ without using $D_{\text{rep}}$ to discriminate from the SC representation perspective does not significantly improve performance.
> >
> > (2) **DFG Inputs**.
> > Following the suggestion from reviewer KQOZ, we conducted ablation studies on the inputs to DFG.
> > The results demonstrate that all inputs are necessary, with the diffusion time $t$ being particularly critical.
> >
> > The completed ablation study has also been updated in the revised PDF.

---

> ### Author Response · Authors · 2024-11-25
> **Looking forward to the discussion**
>
> Dear Reviewer ZpvL,
> We appreciate your constructive feedback provided in your earlier positive review. It indeed inspired us to reflect and improve our paper. As the deadline draws close, could you please kindly let us know if our rebuttal may properly address your concerns? We eagerly look forward to any further thoughts you may wish to share.

---

> > ### Comment · Reviewer_ZpvL · 2024-11-26
> >
> > Thanks for the author's explanation, which effectively addressed my concerns. I have already increased my rating.

---

> > > ### Author Response · Authors · 2024-11-29
> > > **Response to Reviewer ZpvL**
> > >
> > > We sincerely thank you for your positive feedback. We remain open and responsive to any further discussions if you'd like any further information until the conclusion of the discussion stage.

---

### Official Review · Reviewer_Kqoz · 2024-10-30

**Soundness:** 3
**Presentation:** 2
**Contribution:** 3
**Rating:** 6
**Confidence:** 5

**Summary:**

This paper proposes ZeroDiff, a generative framework for zero-shot learning (ZSL). It includes three key components: 1) diffusion augmentation, which transforms limited data into an expanded set of noised data to mitigate generative model overfitting; 2) supervised-contrastive-based representation, which dynamically characterizes semantics of individual samples; and 3) multiple feature discriminators, which evaluate generated features from various perspectives.

**Strengths:**

- The analysis of the performance degradation of ZSL due to a spurious visual-semantic correlation learned from a limited number of seen samples is inspiring.
- The proposed diffusion augmentation and dynamic semantics methods are interesting.

**Weaknesses:**

- Identify and highlight the 1-2 most critical components that provide the key insights. The proposed pipeline is quite complex. It might be hard to tune the whole model. Moreover, it is also hard to know what is the key insight of the proposed method, since there are too many components.
- Provide more motivations and justifications on the design choices of the key components. For example, 1) a clear explanation of the complementary benefits of using both CE and SC loss-based features should be provided. It claims that the SC loss-based representation contains more instance-level semantics. Then why both CE loss-based features and SC loss-based features are used jointly, rather than simply use SC loss-based representation? 2) The theoretical reasoning behind how the denoising discriminator alleviates instance overfitting is required. 3) A detailed justification for each element in the concatenated input to the DFG, explaining how each contributes to the model's performance. In other word, what is the motivation of taking the concatenation of the semantic label a, latent variable z, diffusion time t, noised feature vt, and SC-based representation r0 as condition of DFG? It would be better to give more insightful and in-depth analysis to these design choices, rather than simply verifying by experiments.
- More detailed ablation study would be beneficial to show the effectiveness of the proposed method. Since the proposed method contains many components, the ablation study may not sufficient. For example, though table 3 gives the ablation study of each component, it is still unclear the effectiveness of the detailed design choice within each component, such as the condition of DFG module.
- Provide a more detailed caption for Fig. 4 that explains the key differences between (a), (b), and (c). It is hard to see the key difference among Fig. 4 (a), (b), and (c). It would be better to give more explanations.

**Questions:**

- How to ensure stable training of such a complex pipeline?
- What are the motivations of the design choices of the key modules?
- How to effectively evaluate the design choices of each module?

---

> ### Author Response · Authors · 2024-11-21
> **Official Comment by Authors for Reviewer Kqoz (Part1)**
>
> Thank you for your valuable feedback! We hope that the following clarifications effectively address your concerns.
>
> # W1\&W2\&Q2 (Overall Presentation, Key module motivations):
>
> We apologize for any confusion caused by the complexity of our components and would like to clarify that **our key insights are based on three simple ideas presented in Fig.1.**
>
> Following suggestions from you and reviewer FPAJ, we now provide a step-by-step explanation of our three key insights and their connections to the corresponding modules in our methodology. In this version, readers can seamlessly transition from the baseline f-VAEGAN to our proposed ZeroDiff.
>
> **We have thoroughly refined our manuscript and uploaded the revised PDF, with updates highlighted in $\color{red}{red}$.** You can review these changes in the **revised manuscript PDF** and our summarized **Common response 1**.
>
> We hope these revisions improve readability and effectively address your concerns.
>
> # W2 (CE v.s. SC):
>
> Since SC-based representations exhibit larger intra-class variation compared to CE-based features (Fig. 6 and Fig. 7), they contain more instance-level semantics.
> However, due to the compactness of CE-based features, classification using CE-based features generally outperforms classification using SC-based representations.
> Therefore, utilizing SC-based representations to assist feature generation while classifying in CE space is the optimal approach, compared to using either feature/representation alone for classification.
>
> Fig. 8 includes results from other SOTA methods using SC-based or CE-based features, further demonstrating that CE-based classification generally achieves better performance than SC-based classification.
>
> # W3\&Q3 (More Ablation Study and Module Evaluation):
>
> We have supplemented additional ablation studies in **Common response 2** from two perspectives:
>
> (1) Following the suggestion from reviewer FPAJ, we added results on **more compositions of modules and losses**.
> In addition to the previously mentioned $G+R+D_{\text{adv}}$ and $G+D_{\text{adv}}+D_{\text{diff}}+L_{\text{mu}}$, we also included $G+R+D_{\text{diff}}$, $G+R+D_{\text{adv}}+D_{\text{diff}}$, and $G+R+D_{\text{adv}}+D_{\text{diff}}+L_{\text{mu}}$.
> A new observation from these compositions is that adding $R$ to support $G$, without using $D_{\text{rep}}$ to discriminate from the SC representation perspective, does not significantly improve performance.
>
> (2) Additionally, we followed your suggestion to conduct an ablation study on **DFG inputs**.
> The results indicate that all inputs are necessary, particularly the diffusion time $t$.
>
> The completed ablation study has been updated in the revised PDF.
>
> # W4 (Detailed explanation for Fig. 4):
>
> We find that our SC-based representations exhibit better class discriminability compared to pre-defined semantics and the VADS method, which dynamically updates class-level semantics to instance-level semantics.
> Although VADS can also mine instance-level semantics, it sometimes sacrifices class discriminability.
>
> Specifically, as shown in Fig. 5 in the revised paper (moved from Fig. 4 due to page limitations), we randomly select 8 classes from the CUB dataset and visualize the heatmap of class prototype similarities.
> For the three class similarities marked by the red dash, the similarities from pre-defined class-level semantics are (0.84, 0.82, 0.85), while the similarities produced by VADS remain very close: (0.85, 0.85, 0.86).
> In contrast, our SC-based representations make these hard classes more distinctive, with class similarities of (0.78, 0.89, 0.83).
>
> Due to page limitations and to improve readability, the figure has been moved to the Appendix, while the key findings are retained in the main paper.

---

> > ### Author Response · Authors · 2024-11-21
> > **Official Comment by Authors for Reviewer Kqoz (Part2)**
> >
> > # Q1 (Stable Training):
> >
> > We stabilize our training using two widely adopted techniques from prior research:
> >
> > (1) **Multi-stage Training**.
> > We independently train the extractors $F_{CE}$ and $F_{SC}$, the representation generator $R$, and the feature generator $G$.
> > Once $F_{CE}$ and $F_{SC}$ are trained, they are frozen as $F_{CE}^*$ and $F^*_{SC}$, respectively, allowing us to train $R$ and $G$.
> > In this setup, $R$ and $G$ only need to fit two fixed distributions from $F^*_{SC}$ and $F_{CE}^*$.
> > Previous works, such as f-CLSWGAN, f-VAEGAN, TF-VAEGAN, and VADS, also used separate fine-tuning and generator training stages.
> >
> > (2) **Gradient Penalty (GP)**.
> > The GP term is crucial for stabilizing training. Both of our generators include the GP term (i.e., Eqs. (1, 9, 13) for DFG, and Eqs. (21, 24) for DRG).
> > The GP term penalizes the gradient of the discriminator with respect to the mixed sample $\hat{x} = \alpha x + (1-\alpha) \tilde{x}$.
> > Overly large gradients caused by $\hat{x}$ indicate that discriminator learning outpaces generator learning, potentially leading to exploding or vanishing gradients for the generators.
> > Thus, penalizing the gradient stabilizes GAN training.
> > The GP term is also widely adopted by GAN-based methods such as DSP, VADS, DDGAN, and DiffGAN.

---

> ### Author Response · Authors · 2024-11-25
> **Looking forward to the discussion**
>
> Dear Reviewer Kqoz,
> We appreciate your constructive feedback provided in your earlier positive review. It indeed inspired us to reflect and improve our paper. As the deadline draws close, could you please kindly let us know if our rebuttal may properly address your concerns? We eagerly look forward to any further thoughts you may wish to share.

---

> > ### Comment · Reviewer_Kqoz · 2024-11-26
> >
> > The authors have addressed most of my concerns, and I maintain my original positive rating.

---

> > > ### Author Response · Authors · 2024-11-26
> > > **Response to Reviewer Kqoz**
> > >
> > > We sincerely thank you for your positive feedback.
> > > We remain open and responsive to any further discussions if you'd like any further information until the conclusion of the discussion stage.

---

### Official Review · Reviewer_FpaJ · 2024-10-30

**Soundness:** 4
**Presentation:** 3
**Contribution:** 3
**Rating:** 8
**Confidence:** 4

**Summary:**

This paper proposes to exploit diffusion mechanism to enhance the generative models in zero-shot learning. Specifically, existing generative zero-shot learning methods heavily relies on a large quantity of seen class samples, and the performance of such methods degrades with insufficient samples. To address this problem, the proposed method augments the training set by generating more seen-class and unseen-class samples with diffusion mechanism. With more available samples (either real or generated), the proposed method surpasses existing zero-shot learning methods in both general and sample-limited situations.

Besides, the proposed mutual learning mechanism helps learning better discriminators in the generative process, which in turn improves the generators and classifiers, also contributing to the better performance of the proposed framework.

**Strengths:**

(1)	The proposed method is reasonable and technically solid. To be specific, although addressing the limited data issue by generating more data is quite straightforward, the details of the structure are still novel and effective. For example, the mutual learning mechanism of the discriminators and the incorporation of the diffusion module are well designed.
(2)	The proposed method is effective. As shown in the experiments, the proposed method can achieve better performances than existing methods in most general zero-shot learning situations, and the proposed method consistently surpasses existing generative ZSL methods in data-limited situations. Such results validate the effectiveness of the proposed method.
(3)	The authors provided some visualization results of the learned features, which could provide us with some insights into the possible future direction for further improving ZSL methods.

**Weaknesses:**

(1)	The main manuscript is quite confusing without the appendix. For example, how to finetune the feature extractors, and the details of training and testing are not clearly presented in the main manuscript, making it somehow hard to understand the proposed method when reading the main paper. It would be better if the authors could briefly explain and highlight such key details in the main paper as well as explain them in detail in the appendix.
(2)	It would be better to conduct more experiments in the ablation study part (Table 3). Since the proposed method adopts two kinds of visual features instead of one kind of feature as in most existing ZSL methods, it would be necessary to show how the additional feature contribute to the final performance. That is to say, it would be better to show the performances of G+R+D_{adv}, G+D_{adv}+D_{diff}+L_{mu}, etc..

**Questions:**

(1)	More experiments in the ablation study should be conducted to clearly show the contribution of the additional contrastive feature. Specifically, it would be better to see the results of G+R+D_{adv}, G+D_{adv}+D_{diff}+L_{mu}, and etc..

(2)	In recent years, GPT-series models exhibit amazing performances on zero-shot learning tasks, sometimes even surpasses the models specifically designed for ZSL tasks. It would be interesting to discuss the pros and cons of specifically designed ZSL models compared to such large multi-modal models (LMMs). For example, what are the advantages of specifically designed ZSL models? Are they still necessary in the existence of such large multi-modal models? If so, in which scenarios could the ZSL models perform better than the LMMs? Is it possible to combine both ZSL models and LMMs to achieve even better performances?

It would be sufficient to compare the proposed method with LMMs on a small portion of test samples (e.g. 100 images per dataset), and the results and discussions could be included in a separate new section. I believe these experiments and discussions would surely make the paper more insightful.

**Details Of Ethics Concerns:**

No ethics review needed for this paper.

---

> ### Author Response · Authors · 2024-11-21
> **Official Comment by Authors for Reviewer FpaJ (Part1)**
>
> We thank you for the time and effort you have invested in reviewing this submission and for your positive comments on our work.  We hope the following clarifications effectively address your concerns.
>
> # W1 (Overall Presentation):
>
> **We have thoroughly refined our manuscript and uploaded the revised PDF, with updates highlighted in $\color{red}{red}$.**
> Following the suggestions from you and reviewer KQOZ, we now provide a step-by-step explanation of our three key insights and their connection to the corresponding modules in our methodology.
> In this revised version, readers can seamlessly transition from the baseline f-VAEGAN to our proposed ZeroDiff.
> You can review these changes in the **updated manuscript PDF** and our summarized **Common response 1**.
> We hope these revisions enhance readability and that our clarifications effectively address your concerns.
>
> # W2\&Q1 (More Comprehensive Ablation Study):
>
> The ablation study is also a common consideration.
>
> We have supplemented additional ablation studies in **Common response 2** from two perspectives:
>
> (1) Following your suggestion, we have added results on **more compositions of modules and losses**.
> In addition to the previously mentioned $G+R+D_{\text{adv}}$ and $G+D_{\text{adv}}+D_{\text{diff}}+L_{\text{mu}}$, we have also included $G+R+D_{\text{diff}}$, $G+R+D_{\text{adv}}+D_{\text{diff}}$, and $G+R+D_{\text{adv}}+D_{\text{diff}}+L_{\text{mu}}$.
> A noteworthy observation from these new compositions is that adding $R$ to support $G$, without using $D_{\text{rep}}$ to discriminate from the SC representation perspective, does not significantly improve performance.
>
> (2) Additionally, we followed the suggestion from reviewer KQOZ to conduct an ablation study on **DFG inputs**.
> The results demonstrate that all inputs are necessary, particularly the diffusion time $t$.
>
> The completed ablation study has also been updated in the revised PDF.

---

> ### Author Response · Authors · 2024-11-21
> **Official Comment by Authors for Reviewer FpaJ (Part2)**
>
> # Q2 (LMMs):
>
> Thank you for your insightful comment.
> We acknowledge the impressive performance of GPT-series models and large multi-modal models (LMMs) on zero-shot learning (ZSL) tasks.
> However, specifically designed ZSL models remain relevant for several reasons:
>
> 2-1. **ZSL models v.s. LMMs**:
>
> Existing LMMs including CLIP actually change the definition of zero-shot learning from traditional unseen classes to unseen datasets, which has been clearly discussed in Section 3.2 of the CLIP paper [1]. In other word, the ZSL ability of LMMs is **zero-shot dataset learning**, while our specifically designed ZSL models are still  **zero-shot class learning**. We humbly believe that such difference is fundamental. LMMs suggest a large-scale pre-training hypothesis: only if the pre-trained data are sufficient enough, LMMs can generalize to any unseen datasets or tasks. We argue that this may have four potential issues.
>
> (1) **LMMs may not adhere to the ZSL setting.**
> LMMs generally do not split classes, potentially violating the ZSL premise.
> Although CLIP [1] has provided a comprehensive data overlap analysis in Section 4 stating overlapped data presents minimum influences  on the overall performance, such analysis is not rigorous. The reason is  that this analysis is drawn upon the evaluations on every dataset individually. For example, in CLIP's Appendix 4, the  analysis was made on  a bird dataset, Birdsnap, merely from which overlapped data were detected. It however ignores the fact that  other datasets also contain many bird data, e.g. CIFAR-100. This relaxed evaluation might violate the ZSL setting.
> In contrast, ZSL models explicitly adhere to this ZSL principle, ensuring a more strict evaluation on the zero-shot performance.
>
> (2) **Data used for training certain LMMs  may be limited and/or biased [2].** This may compromise LMMs' zero-shot learning ability. In this case, we still need explore the methods that do not rely on the large-scale pre-training hypothesis, i.e. the specifically designed ZSL methods. In other words, we believe these two different paradigms may be complementary.
>
> (3) **LMMs might perform  worse than ZSL models in many cases.**
>
> In fact, we compare the most basic LMM model CLIP [1] and its derivative CoOp and SHIP in Table 1 of our paper. CLIP performs worse even if it enjoys the huge parameters and training set. Such observations have also been verified in other ZSL works, e.g. ZSLViT [2], VADS [3].
> Besides, ZSL models may perform better when unseen classes are truly novel or in domains requiring strict adherence to the ZSL constraints. Again, in many scenarios, there might be limited data for pre-training. A typical example can be seen in medical domain where existing LMMs work not as well as specifically designed ZSL methods [4].
>
> (4) **Resource-efficient.**
>
> Specifically designed ZSL models are often more resource-efficient, interpretable, and modular, making them suitable for fine-grained or domain-specific tasks, especially in resource-constrained environments.
>
> 2-2. **Combining ZSL Models and LMMs**:
>
> We believe integrating ZSL models with LMMs could leverage the strengths of both approaches, achieving enhanced generalization and performance.
>
> **How LMMs can used to improve ZSL Models**
>
> There are a few ways that LMMs can be used for boosting the performance of ZSL models. On one hand, ZSL Models can take advantages of the powerful representation of LLMs (if the LMM pretraining follows the rigorous class splitting).
> On the other hand, many existing ZSL works can leverage language models to generate class information. For example, I2MVFormer [5] utilizes  GPT-3 that is a fully language model to produce class documents to help ZSL recognition.
> Such a way does not only enjoy the powerful knowledge from LLM, but also guarantees that unseen class images are not leaked.
>
> **How ZSL Models can used to improve LMMs**
>
> Existing ZSL framework can be used to help LLMs to adapt on unseen datasets.
> For example, SHIP just utilized the generative framework f-VAEGAN to help CLIP adapting on many unseen datasets.
>
> In summary, thanks for your suggestions. We are now working on the experiments using LLaMA3 on the test sets of all the  three benchmarks, which is however a bit time-consuming. We will be happy to report these results in our revision once we have any progress.
>
> [1] Learning transferable visual models from natural language supervision, ICML, 2021.
>
> [2] Progressive Semantic-Guided Vision Transformer for Zero-Shot Learning, CVPR, 2024.
>
> [3] Visual-Augmented Dynamic Semantic Prototype for Generative Zero-Shot Learning, CVPR, 2024.
>
> [4] Generative multi-label zero-shot learning, IEEE TPAMI, 2023.
>
> [5] I2mvformer: Large language model generated multi-view document supervision for zero-shot image classification, CVPR, 2023.

---

> > ### Comment · Reviewer_FpaJ · 2024-11-22
> >
> > I appreciate the authors for the insightful discussions about the relations between LMMs and ZSL methods. I agree with the authors that it is hard to know whether the samples of a certain class is presented in the pre-training dataset of LMMs, making it challenging to merge LMMs into ZSL methods under the current ZSL settings. Still, I believe that combining LMMs/LLMs with ZSL methods could be a promising way in the future. To this end, new datasets suitable for evaluating such combined methods are necessary, and more creative ways to combine them might be beneficial.

---

> > > ### Author Response · Authors · 2024-11-24
> > > **Response for Reviewer FpaJ**
> > >
> > > Thank you for your thoughtful feedback and for acknowledging the challenges of integrating LMMs/LLMs into ZSL under current settings. We fully agree that combining these approaches holds great potential. Developing new datasets and innovative methods for such evaluations will not only advance the field but also help guide our future research directions in exploring the synergy between LMMs/LLMs and ZSL methods.

---

### Author Response · Authors · 2024-11-21
**Response to All Reviewers (Part 1)**

We sincerely thank all the reviewers for their valuable feedback. To address your concerns and questions, we have provided responses to two common points of consideration, along with detailed, individualized replies to each reviewer. We hope these explanations offer a clearer and more comprehensive understanding of our work.

# Common consideration1: Overall Presentation

We apologize for any confusion caused by the lack of clarity in our presentation.

We would like to clarify that, **despite the complexity of our components, the core insights of our work remain the three simple ideas presented in Figure 1**. To enhance the clarity of our presentation, we have revised the description of our method. Based on the suggestions from reviewers FPAJ and KQOZ, we now provide a step-by-step explanation of the three key insights and their connection to the corresponding modules in our methodology. In the updated version, readers can seamlessly transition from the baseline f-VAEGAN to our proposed ZeroDiff.
**We have uploaded the revised paper PDF, with changes highlighted in $\color{red}{red}$ for easy reference.**
We hope these revisions improve the readability and understanding of our work.

For convenience, we have also summarized the key motivations behind our insights here.

**Key insight 1: diffusion augmentation.**

One of the reasons for spurious visual-semantic correlations is that discriminators tend to memorize the limited training data.
To address this issue, our first key insight is to leverage the diffusion mechanism to expand the limited training set into an infinite pool of noised data, thereby mitigating this problem.
Specifically, we enable the generator $G$ to denoise (i.e., generate) clean fake features from noised real samples. This is achieved by introducing diffusion time $t$ and corresponding noised samples $v_t$ into the input of $G$ along with a diffusion discriminator $D_{diff}$ to evaluate whether the generated samples align with the diffusion process.

**Key insight 2: SC-based representations.**

Another cause of spurious visual-semantic correlation is the use of class-level semantic labels $\mathbf{a}$, where all instances within a class share the same label. This can lead to mismatched correlations between instances and their semantics. For example, as shown in Fig. 1, all images of the ``fox'' class are labeled with the semantics [red] and [white], even though white foxes are not [red]. Such mismatches amplify spurious visual-semantic correlations.

From previous literature, we observe that SC loss can increase intra-class variation, suggesting its ability to model instance-level semantics. Our experiments, shown in Fig. 6 and Fig. 7, support this finding. For instance, different sub-classes (e.g., red fox, white fox, and gray fox) within the ``fox'' class are grouped separately in SC-based space.

Building on this idea, we utilize a frozen representation extractor $F_{SC}^*$ to extract SC-based representations $r_{0}$, which are then used as input to the generator $G$ to provide instance-level semantics. To ensure consistency between the generated features $\mathbf{r}^s_{0}$ and the SC-based representations $\mathbf{r}^s_{0}$, we introduce a representation discriminator $D_{rep}$ to assess whether the generated samples align with the input representations.

It is important to note that during the testing stage, real unseen class representations $\mathbf{r}^{u}_{0}$ are unavailable. To address this, we train a Diffusion-based Representation Generator (DRG) $R$ to learn the mapping from class-level semantic labels $\mathbf{a}$ to instance-level SC representations $\mathbf{r}_0$. The DRG training process is similar to that of DFG, and detailed training algorithms are provided in Algorithm 2.

**Key insight 3: Mutual-learned discriminators.**

Since our three discriminators evaluate features in different ways, they use distinct criteria for their judgments. By enabling them to learn from each other, we can create stronger discriminators, which in turn provide better guidance for the generator. To achieve this, we propose $L_{mu}$, which aligns the Wasserstein distance of the discriminators with a Noise-to-Data (N2D) ratio. This ratio controls the alignment strength at different diffusion times.

---

> ### Author Response · Authors · 2024-11-21
> **Response to All Reviewers (Part 2)**
>
> # Common consideration2: Ablation study
>
> Our ablation study is also a common consideration.
> We have supplemented additional ablation studies from two perspectives.
> For convenience, we also provide the results and analysis with the openreview fashion here.
>
> (1) Following the suggestion from reviewer FPAJ, we have supplemented results concerning **additional compositions of modules and losses**, including $G+R+D_{\text{adv}}$, $G+D_{\text{adv}}+D_{\text{diff}}+L_{mu}$, $G+R+D_{\text{diff}}$, $G+R+D_{\text{adv}}+D_{\text{diff}}$, and $G+R+D_{\text{adv}}+D_{\text{diff}}+L_{\text{mu}}$.
>
> | ID | $G$ | $R$ | $D_{adv}$ | $D_{diff}$ | $D_{rep}$ |  $\mathcal{L}_{mu}$ |  AWA2 T1 | AWA2 H | CUB T1 | CUB H| SUN T1 | SUN H |
> | - | :-: | :-: | :-: |  :-: | :-: |  :-: | :-: | :-: | :-: | :-: | :-: | :-: |
> | a | ✓ | × | ✓ | × | × | × | 79.9 | 67.0 | 83.7 | 78.5 | 74.7 | 57.8 |
> | b | ✓ | × | ✓ | ✓ | × | × | 82.3 | 72.9 | 84.1 | 79.5 | 75.1 | 58.7 |
> | $\color{crimson}{c}$ | $\color{crimson}{✓}$ | $\color{crimson}{×}$  | $\color{crimson}{✓}$ | $\color{crimson}{✓}$ |        $\color{crimson}{×}$ | $\color{crimson}{✓}$ | $\color{crimson}{82.7}$ | $\color{crimson}{73.3}$ | $\color{crimson}{84.7}$ | $\color{crimson}{80.9}$ | $\color{crimson}{75.4}$ | $\color{crimson}{59.0}$ |
> | d | × | ✓ | ⚪ | ⚪ | × | × | 83.8 | 67.1 | 78.3 | 66.8 | 71.6 | 47.6 |
> | $\color{crimson}{e}$ | $\color{crimson}{✓}$ | $\color{crimson}{✓}$ | $\color{crimson}{✓}$ | $\color{crimson}{×}$ | $\color{crimson}{×}$ | $\color{crimson}{×}$ | $\color{crimson}{79.3}$ | $\color{crimson}{72.8}$ | $\color{crimson}{83.4}$ | $\color{crimson}{78.5}$ | $\color{crimson}{73.6}$ | $\color{crimson}{56.2}$ |
> | $\color{crimson}{f}$ | $\color{crimson}{✓}$ |  $\color{crimson}{✓}$ |  $\color{crimson}{×}$ | $\color{crimson}{✓}$ | $\color{crimson}{×}$ | $\color{crimson}{×}$ | $\color{crimson}{80.3}$ | $\color{crimson}{69.0}$ | $\color{crimson}{85.8}$ | $\color{crimson}{80.5}$ | $\color{crimson}{73.5}$ | $\color{crimson}{56.4}$ |
> | $\color{crimson}{g}$ | $\color{crimson}{✓}$ | $\color{crimson}{✓}$ | $\color{crimson}{✓}$ | $\color{crimson}{✓}$ | $\color{crimson}{×}$ | $\color{crimson}{×}$ | $\color{crimson}{82.9}$ | $\color{crimson}{73.4}$ | $\color{crimson}{84.8}$ | $\color{crimson}{79.1}$ | $\color{crimson}{74.4}$ | $\color{crimson}{57.0}$ |
> | $\color{crimson}{h}$ | $\color{crimson}{✓}$ | $\color{crimson}{✓}$ | $\color{crimson}{✓}$ | $\color{crimson}{✓}$ | $\color{crimson}{×}$ | $\color{crimson}{✓}$ | $\color{crimson}{85.2}$ | $\color{crimson}{78.3}$ | $\color{crimson}{87.2}$ | $\color{crimson}{81.2}$ | $\color{crimson}{76.8}$ | $\color{crimson}{59.4}$ |
> | i | ✓ | ✓ | ✓ | ✓ | ✓ | × | 84.6 | 77.5 | 84.8 | 79.3 | 76.8 | 59.2 |
> | j | ✓ | ✓ | ✓ | ✓ | ✓ | ✓ | 87.3 | 81.4 | 87.5 | 81.6 | 77.3 | 59.8 |
> Table 1. Ablation study of our ZeroDiff. ✓ and × denote yes and no. ⚪ denotes using corresponding versions for DRG.
>
>
> From the results, we can draw three key observations:
>
> (i) Comparing pairs (b,c), (g,h), and (i,j), we observe that $D_{\text{diff}}$ and $L_{\text{mu}}$ consistently improve performance across the three datasets.
>
> (ii) **(additional) Comparing the pairs (a,e) and (b,g), we find that adding $R$ to support $G$, without using $D_{\text{rep}}$ to discriminate from the SC representation view, does not significantly improve performance.**
>
> (iii) However, once both $R$ and $D_{\text{rep}}$ are used, the performance improvements become substantial, as seen in pairs (g,i) and (h,j). This indicates that $G$ benefits significantly from the inclusion of both $R$ and $D_{\text{rep}}$ in many cases. It also supports our motivation that SC-based representations can capture instance-level semantics to aid feature generation.

---

> > ### Author Response · Authors · 2024-11-21
> > **Response to All Reviewers (Part 3)**
> >
> > (2) Additionally, following the suggestion from reviewer KQOZ, we have included another ablation study regarding **the input to $G$**. The results show that all inputs are necessary, especially the diffusion time $t$.
> >
> > | ID | $\mathbf{a}$ | $\mathbf{r}_0$ | $\mathbf{v}_t$ | $t$ |  AWA2 T1 | AWA2 H | CUB T1 | CUB H| SUN T1 | SUN H |
> > | - | :-: | :-: |  :-: | :-: | :-: | :-: | :-: | :-: | :-: | :-: |
> > | $\color{crimson}{a}$ | $\color{crimson}{✓}$ | $\color{crimson}{×}$ | $\color{crimson}{×}$ | $\color{crimson}{×}$ | $\color{crimson}{77.0}$ | $\color{crimson}{69.8}$ | $\color{crimson}{85.5}$ | $\color{crimson}{80.0}$ | $\color{crimson}{76.8}$ | $\color{crimson}{58.3}$ |
> > | $\color{crimson}{b}$ | $\color{crimson}{×}$ | $\color{crimson}{✓}$ | $\color{crimson}{×}$ | $\color{crimson}{×}$ | $\color{crimson}{80.0}$ | $\color{crimson}{75.7}$ | $\color{crimson}{85.1}$ | $\color{crimson}{79.9}$ | $\color{crimson}{73.1}$ | $\color{crimson}{55.9}$ |
> > | $\color{crimson}{c}$ | $\color{crimson}{✓}$ | $\color{crimson}{✓}$ | $\color{crimson}{×}$ | $\color{crimson}{×}$ | $\color{crimson}{84.3}$ | $\color{crimson}{76.0}$ | $\color{crimson}{86.9}$ | $\color{crimson}{80.6}$ | $\color{crimson}{77.0}$ | $\color{crimson}{58.5}$ |
> > | $\color{crimson}{d}$ | $\color{crimson}{✓}$ | $\color{crimson}{✓}$ | $\color{crimson}{✓}$ | $\color{crimson}{×}$ | $\color{crimson}{84.2}$ | $\color{crimson}{78.1}$ | $\color{crimson}{85.5}$ | $\color{crimson}{80.3}$ | $\color{crimson}{75.0}$ | $\color{crimson}{57.6}$ |
> > | e | ✓ | ✓ | ✓ | ✓ | 87.3 | 81.4 | 87.5 | 81.6 | 77.3 | 59.8 |
> > Table2. Ablation study of the input of our DFG. ✓ and × denote yes and no.
> >
> > Specifically:
> > (i) IDs a, b, and c confirm that class-level semantic labels $\mathbf{a}$ and instance-level SC representations $\mathbf{r}_0$ consistently improve performance across the three datasets.
> > (ii) IDs c and d show that if only $\mathbf{v}_t$ is used, our method does not significantly benefit from reconstructing $\tilde{\mathbf{v}}_0$ from $\mathbf{v}_t$, indicating that the diffusion time $t$ is a critical input for our DFG $G$.
> > (iii) IDs d and e illustrate that inputting both $\mathbf{v}_t$ and $t$ results in a significant improvement, demonstrating the effectiveness of our diffusion augmentation.

---

> > ### Comment · Reviewer_FpaJ · 2024-11-22
> >
> > My concerns about the ablation study have been addressed. It is interesting to see that the R component only works well together with D_{rep}, suggesting that simply adding more features might not be helpful for ZSL methods. However, such a phenomenon seems strange, since more information are provided to the model, and we would natrually expect the model to perform better with more features. It would be better if the authors could provide a possible explanation to this phenomenon.

---

> > > ### Author Response · Authors · 2024-11-24
> > > **Response for Reviewer FpaJ**
> > >
> > > Thanks for your timely response.
> > > We indeed appreciate the chance to communicate with you, which could help further clarify our method.
> > >
> > > Using $R$ without $D_{\text{rep}}$ may not consistently improve performance in some cases, as seen in ID a vs. ID e.
> > > This is primarily because $D_{rep}$ helps the generator $G$ establish a correct relationship between CE-based and SC-based features, which represent instance-level semantics.
> > > This relationship is particularly critical during testing, where the fake instance-level semantics generated by $R$ are fed into $G$ to produce features for unseen classes.
> > > Without the correct relationship provided by our $D_{rep}$, $G$ might amplify the negative impact caused by suboptimal features from $R$.
> > >
> > > A similar observation can be made from the additional ablation results in Table 2 of the common response (Part 3).
> > > If diffusion time $t$ is not used to guide $G$, the generator fails to learn the correct diffusion process.
> > > This is because the variation of the diffused features $\mathbf{v}_t$ becomes dramatic as $t$ changes, and without the hint from $t$, $G$ struggles to capture this process effectively.
> > >
> > > We thank you again for the inspiring comments. We hope our response could resolve your concerns and look forward to communicating with you if you may have any further suggestions.

---

### Meta-Review · Area_Chair_9CWc · 2024-12-20

**Metareview:**

# Summary and Recommendation for Acceptance

---
## **Strengths**
1. **Novel Contributions**:
   - Proposes **ZeroDiff**, a generative framework addressing spurious visual-semantic correlations in zero-shot learning (ZSL). Key innovations include:
     - **Diffusion augmentation**: Expands limited datasets to mitigate overfitting.
     - **Supervised-contrastive representations**: Captures dynamic sample-level semantics.
     - **Mutual-learned discriminators**: Evaluates features from multiple perspectives, enhancing generation and classification performance.
   - Empirical demonstration of robust performance with only 10% of training data.

2. **Experimental Rigor**:
   - Extensive evaluations on popular benchmarks (AWA2, CUB, and SUN) showing superior performance in general and data-limited scenarios.
   - Comprehensive ablation studies validate the effectiveness of individual components.

3. **Theoretical Insights**:
   - Identifies and addresses the degradation in generative ZSL methods caused by spurious correlations.
   - Provides sound theoretical motivations for the proposed design.

4. **Practical Relevance**:
   - Framework is particularly suited for resource-constrained settings due to its data efficiency.

5. **Community Contribution**:
   - Commitment to open-sourcing the code for reproducibility and further exploration by the community.

---

## **Weaknesses**
1. **Complexity**:
   - The model is intricate, with multiple components that complicate interpretation, training, and tuning, potentially deterring adoption.

2. **Presentation Issues**:
   - Initial manuscript was unclear, requiring heavy reliance on the appendix for understanding.
   - Figures (e.g., Fig. 4) lacked sufficient captions to highlight key distinctions.

3. **Incomplete Baseline Comparisons**:
   - Concerns about fairness in baseline comparisons due to differences in fine-tuning methods.

4. **Overlapping Features**:
   - Lack of initial clarity on the complementary benefits of combining SC-based and CE-based features.

---

## **Authors' Mitigation**
1. **Complexity and Presentation**:
   - Revised the manuscript with a clearer, step-by-step explanation of methodology and improved figures.
   - Provided detailed justifications for design choices, particularly the combination of SC and CE features.

2. **Baseline Comparisons**:
   - Clarified the use of fine-tuned features across methods and conducted additional ablation studies to ensure fairness.

3. **Feature Redundancy**:
   - Explained the complementary roles of SC and CE features:
     - **SC** captures instance-level semantics.
     - **CE** provides compact and effective classification performance.
   - Empirical evidence supporting this claim was presented in supplementary analyses.

4. **Evaluation of Design Choices**:
   - Expanded ablation studies to assess the contribution of individual components and inputs (e.g., diffusion time).

---

## **Remaining Weaknesses**
1. **Complexity**:
   - Despite revisions, the framework remains complex and challenging to train, which may limit its accessibility to practitioners.

2. **Baseline Coverage**:
   - While outperforming generative ZSL models, further comparisons to large multimodal models (e.g., CLIP) were not fully addressed, leaving opportunities for future work.

---

## **Justification for Acceptance**
This paper effectively addresses a significant limitation in ZSL by mitigating spurious visual-semantic correlations in limited data scenarios. Its novel contributions, including diffusion augmentation and supervised-contrastive representations, advance the state of the art. The rigorous empirical validation and comprehensive rebuttals to reviewers' concerns further bolster its credibility.

While challenges related to complexity and broader comparisons persist, these are outweighed by the substantial theoretical and practical contributions. Given its strong performance and potential impact on ZSL research, I recommend acceptance.

**Additional Comments On Reviewer Discussion:**

Please refer to details in the above section.

---

### Decision · Program_Chairs · 2025-01-22

Accept (Poster)